# Efficient encoding of large antigenic spaces by epitope prioritization with Dolphyn

Anna-Maria Liebhoff [1,2], Thiagarajan Venkataraman[2], William R. Morgenlander [2], Miso Na [2], Tomasz Kula [3,4], Kathleen Waugh[5], Charles Morrison[6], Marian Rewers [5], Randy Longman[7], June Round [8], Stephen Elledge [3,4], Ingo Ruczinski[9], Ben Langmead[1] & H. Benjamin Larman [2] ✉

We investigate a relatively underexplored component of the gut-immune axis by profiling the antibody response to gut phages using Phage Immunoprecipitation Sequencing (PhIP-Seq). To cover large antigenic spaces, we develop Dolphyn, a method that uses machine learning to select peptides from protein sets and compresses the proteome through epitope-stitching. Dolphyn compresses the size of a peptide library by 78% compared to traditional tiling, increasing the antibody-reactive peptides from 10% to 31%. We find that the immune system develops antibodies to human gut bacteria-infecting viruses, particularly *E.coli*-infecting *Myoviridae*. Cost-effective PhIP-Seq libraries designed with Dolphyn enable the assessment of a wider range of proteins in a single experiment, thus facilitating the study of the gut-immune axis.

The human gut microbiome is a critical determinant of human health. However, the mechanisms underlying the interactions between the host and the diverse microorganisms in the gut, including bacteria, fungi, phages, archaea, and other members of the microbiota, remain largely unknown. Gut phages, which infect bacteria in the gut, are increasingly recognized as important contributors to the host-microbe-immune axis. These bacterial viruses have been described as the "puppet masters" of gut bacteria[1]. The immune system, which defends against foreign invaders and protects tissue homeostasis, plays a major role in gut related health. Host antibodies may, for instance, directly impact the composition of the bacterial population in the gut by neutralizing specific gut phages. Recent data[2] support the presence of immune responses to gut phages, along with the importance of anti-gut phage antibodies in inflammatory bowel disease[3]. However, exhaustively characterizing the immune response to gut phages requires testing a very large number of potential antigenic targets. Camarillo-Guerrero

et al., recently published a database of gut phage sequences, most of which have not been previously described[1]. Though it is smaller than that of gut bacteria, the gut phage metaproteome is still too vast to be represented via oligonucleotide library synthesis.

Phage ImmunoPrecipitation Sequencing (PhIP-Seq) is a technique for profiling the reactivity of an individual's antibody repertoire to a wide range of antigens. Previous publications demonstrate the concordance between PhIP-Seq data and peptide ELISA data[4,5]. This technique involves designing peptides that tile across proteins, synthesizing oligonucleotide libraries that encode the peptides, and cloning the oligo libraries into a phage display vector. Phage display and immunoprecipitation are used to test serum samples for antibody binding to all peptides in parallel, since DNA sequencing is used to determine the relative abundance of the immunoprecipitated population. Reference protein sequences from public databases typically serve as the basis for these phage display libraries that normally consist

[1]Department of Computer Science, Johns Hopkins University, Baltimore, MD, USA. [2]Institute of Cell Engineering, Division of Immunology, Department of Pathology, Johns Hopkins University, Baltimore, MD, USA. [3]Department of Genetics, Harvard Medical School, Boston, MA, USA. [4]Division of Genetics, Department of Medicine, Howard Hughes Medical Institute, Brigham and Women's Hospital, Boston, MA, USA. [5]Barbara Davis Center for Diabetes, University of Colorado Denver, Aurora, CO, USA. [6]Behavioral, Clinical and Epidemiologic Sciences, FHI 360, Durham, NC, USA. [7]Jill Roberts Institute for Research in IBD, Division of Gastroenterology and Hepatology, Department of Medicine, Weill Cornell Medicine, New York, NY, USA. [8]Department of Pathology, Division of Microbiology and Immunology, University of Utah School of Medicine, Salt Lake City, UT, USA. [9]Department of Biostatistics, Johns Hopkins University, Baltimore, MD, USA. ✉e-mail: hlarman1@jhmi.edu

of 56–90 amino acid long peptides. PhIP-Seq libraries have been designed to span the human proteome[6], common viruses[7], allergens[8,9], selected gut bacteria[10], and protein toxins[11], providing insights into health and disease.

To date, PhIP-Seq libraries have been designed primarily using Pepsyn, a software tool that performs uniform peptide tiling across proteins[12]. Typically, Pepsyn is used to generate peptides that overlap by half the tile size, in order to prevent disrupting epitopes. This approach results in roughly 2× coverage of the input proteome. Representing the gut phage proteome in this manner would be intractable, which motivated us to develop a new method for creating more efficient representations of large antigenic spaces.

We reasoned that an efficient peptide library would selectively include antibody epitopes that tend to be targeted by the human immune system (public epitopes). Since the study of gut phages remains underdeveloped, their proteins are largely absent from databases such as the Immune Epitope Database[13]. However, it has been recently reported that public epitopes tend to contain amino acid sequence features that are important for interactions with germline-encoded antibody domains[14]. This suggests the potential to identify peptide sequences that are more likely to contain public epitopes by their amino acid composition. Indeed, previous studies[15,16] have also developed models to achieve this goal.

Our compact library design method, named Dolphyn, employs two components. The first is a binary machine-learning classifier of 15 amino acid peptides that was trained on our large public epitope reactivity database. The second component is a new strategy for combining multiple regions of a protein into one peptide, for simultaneous testing within a single synthesized oligo. To demonstrate its utility, we used Dolphyn to create a PhIP-Seq peptide library and profiled gut phage proteome antibodies of healthy individuals.

## Results

**Tiling peptide libraries tend to contain only a small proportion of reactive peptides.** Figure 1A shows the proportion of reactive peptides, with the highest proportion (15.4% per 100 individuals) found in the VirScan library[7]. The low fraction of reactive peptides, even in libraries dominated by human pathogens, highlights that new library designs could be made significantly more efficient.

We propose a new method for designing PhIP-Seq libraries for proteomes that are too large to tile exhaustively with Pepsyn. Our method effectively compresses the PhIP-Seq library to a practical size, while minimizing lost sensitivity to detect protein-level reactivity.

In the following subsections, we analyze epitope characteristics, with a particular emphasis on epitope length and predictive patterns. We then formulate a binary classifier to predict presence of epitopes within 15-amino acid peptides. Next, we introduce the Dolphyn algorithm, which efficiently compresses large sets of proteins into concise peptide libraries. To validate the algorithm, we compare a conventional library design with one produced by Dolphyn. Finally, we

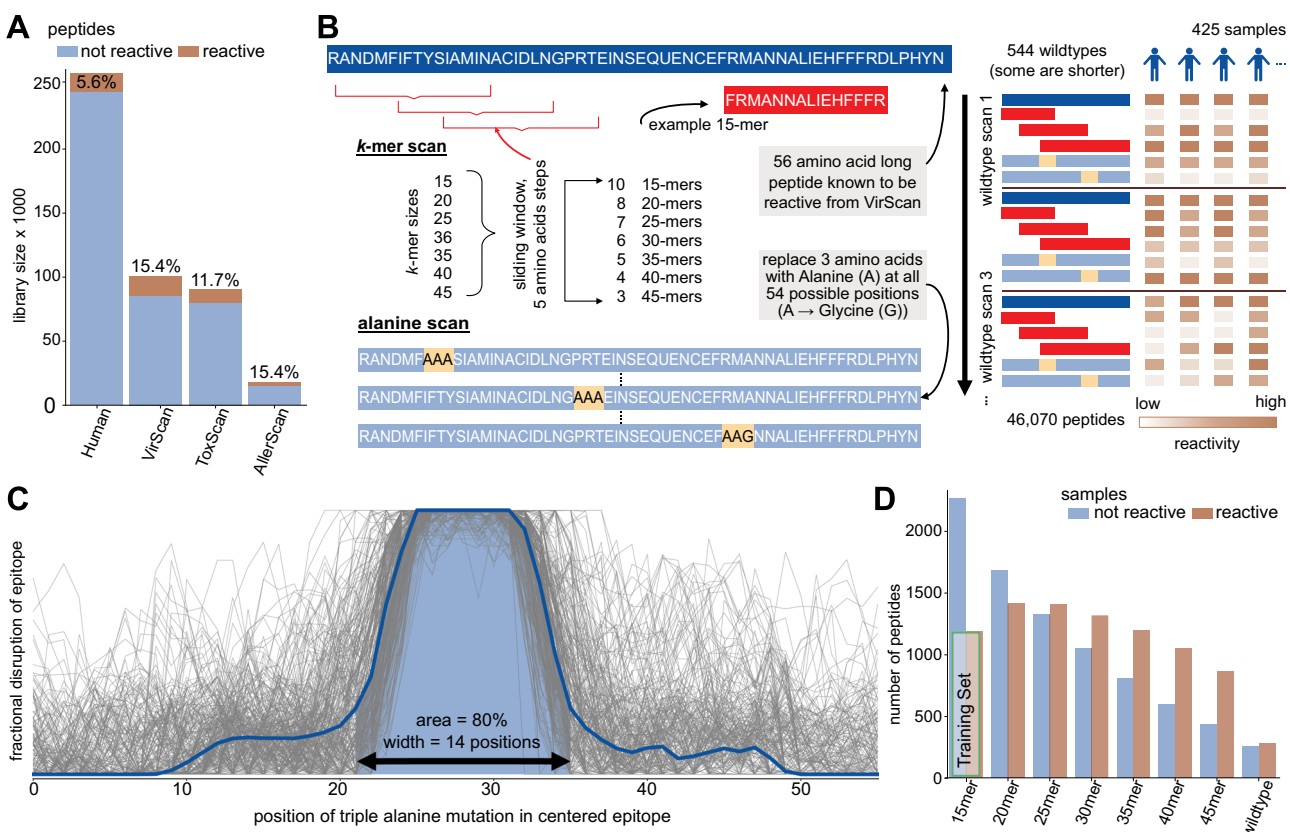

**Fig. 1 | Antibody epitope analysis using programmable phage display of peptide libraries. A** Complexity and reactivity of previously published peptide libraries. Bars show number of peptides included in each library and the percentage of peptides that are reactive in at least 1% of samples from the Vaccine Research Center (VRC) cohort. **B** The Public Epitope Data Set includes a k-mer scan and an alanine scan of 544 virus-derived 56 amino acid long immunodominant peptides in 59 individuals (425 samples). The k-mer scan consists of k = 15–45 amino-acid-long sub-peptides of the wildtypes, starting every 5 amino acids. The alanine scan consists of modified versions of the wildtype peptide where triplets of amino acids were replaced with three alanines. Wildtype alanines are replaced by glycines. **C** Compilation of alanine scans from reactive peptides and individuals. Each gray line is the difference of the alanine peptide at that position to the wildtype reactivity in one individual. Only lines indicating a single epitope were included and shifted to the center. Blue line indicates median. **D** Summarized k-mer scans. A peptide is considered reactive if more than one percent of samples react to it. The Training Set indicates those peptides used in the prediction model introduced in Fig. 2. | Source data are provided as a Source Data file.

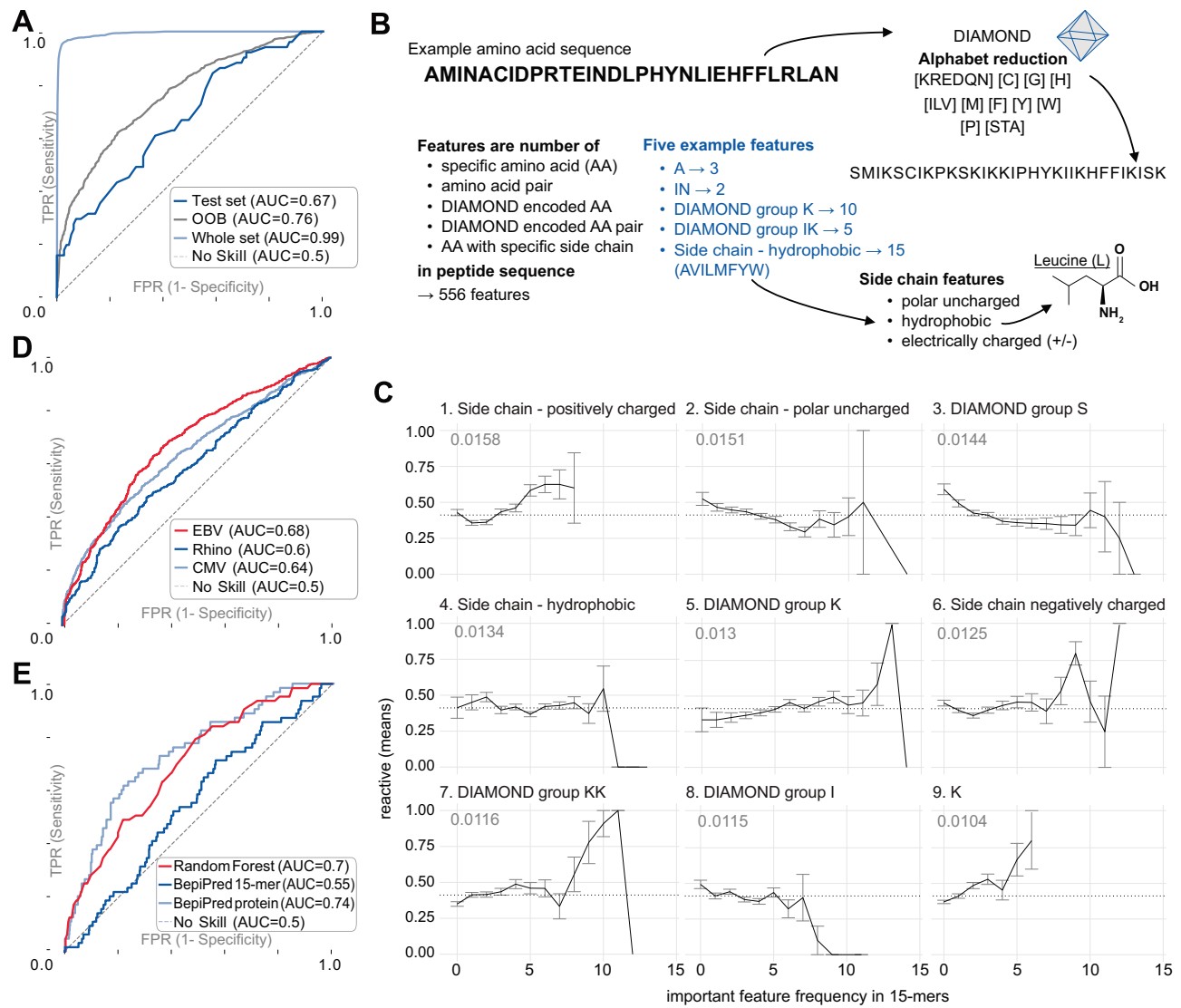

**Fig. 2 | Binary classification of 15-mer peptides containing an epitope. A** ROC curves for random test and training sets and random forest out-of-bag (OOB) set. 5% of the training data was split off as a test set, with no overlap of wildtypes between the sets. **B** Various peptide features were used in a random forest model, such as the frequency of natural and DIAMOND encoded amino acids (AA) and AA-pairs, and AA-side chain properties; 556 features were included in total. **C** Top nine most important features in random forest model. The x-axis indicates the frequency of the AA sequence feature (max 15/15) and the y-axis shows the proportion and standard error of reactive peptides in individuals in the VRC cohort. The dotted line indicates the reactivity ratio of all 3456 peptides in the dataset. The number indicates the mean decrease in impurity, an importance score of a feature. **D** ROC curves on 2263 independent 56-mer Epstein-Barr virus (EBV) peptides, 945 rhinovirus peptides and 4016 human cytomegalovirus (CMV) peptides, tested on the VRC cohort. **E** ROC curve of 15-mer peptides in an Enterovirus peptide library, tested on the DAISY cohort, for random forest and BepiPred-3.0 predictions. **A, D, E** TPR/FPR is true/false positive rate, respectively.| Source data are provided as a Source Data file.

construct and evaluate a pilot library encompassing gut phage proteins, employing and comparing the traditional and novel library design methods.

## Epitopes contained in reactive peptides

To identify biochemical features associated with public epitopes, we selected highly immunogenic peptides (wildtype) for examination in a cohort of 59 individuals (425 samples). The resulting Public Epitope Data Set elucidates the fine specificities of antibody epitope selection using two types of sub-peptides, the *alanine scan* and the *k-mer scan* (Methods and Fig. 1B, data available).

The alanine scan library of peptides contains a series of peptides identical to each of the native peptide sequence but with triple alanine substitutions scanning from N- to C-terminus of each peptide. Triple alanine substitutions can interrupt antibody binding and thus reveal

the precise determinants of an epitope within a longer peptide sequence. Figure 1C normalizes, centers, and overlays this information for all individuals with reactivity to a native public epitope. 80% of the mean reactivity curve spans 14 amino acid positions, suggesting that most linear public epitopes can be captured by peptides of this length.

Figure 1D presents the results of the $k$-mer scan to assess reactivity for various peptide lengths. The $k$-mer scan considers sub-peptides of varying lengths ($k = 15$–45 amino acids) derived from the wildtype sequence. They are tiled in steps of five amino acids along the sequence. As expected, the number of non-reactive peptides increased with shorter peptide lengths. Wildtype peptides that are reactive may contain one or more epitopes, whereas the shorter sub-peptides derived from the wildtype peptides are likely to contain only one epitope. Shorter peptides are also unlikely to contain many excess amino acids outside of a reactive epitope, as compared to the 56 amino

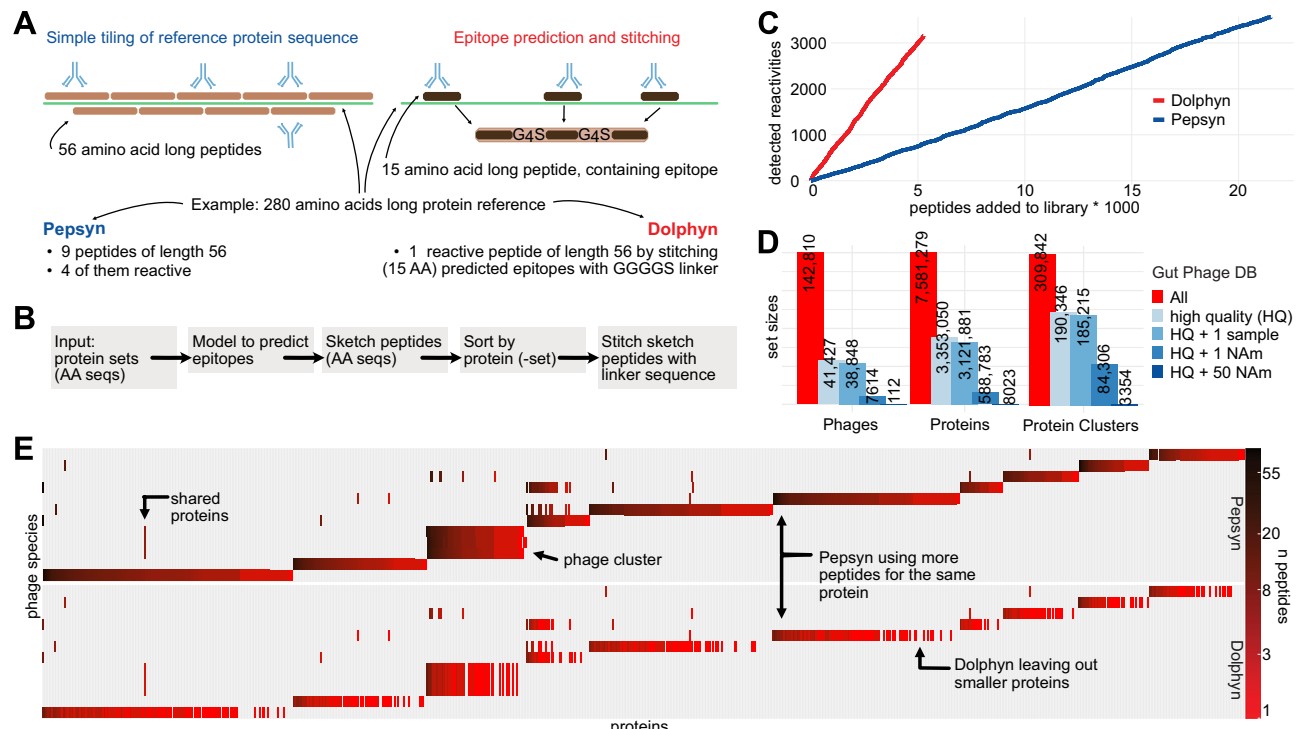

**Fig. 3 | The Dolphyn library design algorithm. A** Pepsyn tiles the protein sequence uniformly with fixed size, overlapping peptides. Only a fraction of the peptides is reactive to antibodies. Dolphyn selects a smaller number of short peptides per protein, favoring peptides more likely to be epitopes and stitching them into one composite tile using G4S-linkers to separate the epitopes. **B** Dolphyn Workflow. The package includes modularized steps for processing the amino acid sequences (AA seqs), e.g. including the random forest model (Fig. 2). **C** Cost-effectiveness of library design. As more peptides are included, i.e., synthesized and sequenced (x-axis), more immune responses were detected in the test cohort (y-axis). Dolphyn peptides are ordered by their mean prediction value of the contained epitopes, from highest to lowest. Pepsyn peptides are ordered randomly. **D** The metadata analysis of Gut Phage Database-included phages, proteins and protein clusters reveals sub-sets of genome quality and abundance in North American (NAm) metagenomic samples. **E** Library composition with Pepsyn versus Dolphyn. 12 example phage species (y-axis) were represented by Pepsyn or Dolphyn. Proteins are left out when fewer than three epitopes are predicted. | Source data are provided as a Source Data file.

acid long peptides. 15-mer peptides were therefore selected to serve as a dataset for training a binary classifier to predict whether or not a peptide includes an epitope.

## Binary classification of epitope peptides

We trained a random forest classifier using peptides that are reactive in many individuals and an equally sized set of peptides that are not reactive in the Public Epitope Data Set cohort. Our model and training data are freely available on GitHub. We provide models for peptides of amino acid lengths 15–45.

Figure 2A shows that our classifier fits the data well (Area under the ROC Curve (AUC) 0.99 on the whole dataset, training set was 95%), with the test-set (5%) AUC of 0.67. The higher out-of-bag AUC of 0.76 is likely due to existing peptide sequence similarity in the training data, while the test set was selected to avoid sequence similarity to better measure generalized prediction capability.

The 556 features used for the model (Fig. 2B) included frequencies of amino acids, amino acid pairs, as well as frequencies of classes of amino acids defined by the DIAMOND alphabet reduction[17] and side-chain type. We found that amino acid frequencies were informative regardless of whether the model included their positions along the peptide. Furthermore, we explored using protein structures predicted by BepiPred-2.0[18] and RePROF[19] as features in our model. Including these models provided insufficient enhancement of classification performance to justify the additional computational costs associated with extracting and incorporating these features.

We discovered the most important features in predicting if a 15-mer contained an epitope to be the amino-acid side-chain

frequencies. All four side-chain types were among the nine most important features (Fig. 2C). The number of positively charged amino acids in the peptide was most crucial. For instance, Fig. 2C illustrates that if a 15-mer peptide contains between 5 to 8 amino acids with a positively charged side chain, it has a higher likelihood of containing an epitope.

The frequency of amino acids in the DIAMOND serine group (including threonine and alanine) is the most important feature. Lysine (K) frequency is also important, as is its DIAMOND group (including arginine, glutamic acid, aspartic acid, glutamine and asparagine), and K-DIAMOND pairs. Interestingly, a recent study also found lysine to be an important feature of epitopes. A germline encoded feature of antibodies called the "GRAB" motif was described as playing an important role in recognizing public epitopes[14]. In humans, these epitopes enrich lysines on their borders if recognized by antibodies using a lambda light chain.

To assess performance on an independent dataset, we collected a set of Epstein-Barr virus (EBV) peptides, rhinovirus peptides and human cytomegalovirus peptides that had been previously screened in the VirScan study by Xu et al.[7]. Given the high prevalence of rhinovirus, cytomegalovirus and EBV infection worldwide[20], these datasets provided valuable ground-truth for evaluating the presence of epitopes in these peptides. Since the VirScan peptides are 56 amino acids long, whereas our random forest model was trained on 15-mers, we evaluated all possible 15-mers within the 56-mer peptides and used their mean to generate the ROC curve shown in Fig. 2D. Despite the necessary transformation to adopt the random forest model, it predicted antibody epitopes of EBV peptides with an AUC of 0.68.

**BOX 1**

# Dolphyn algorithm pseudo code

```
for each protein_sequence:

//PREDICTION STEP

    for each 15mer in 15-amino-acid-sub-peptide:

probabilities[15mer] = contains_epitope(15mer) // C1
threshold = 0.5
epitope_15mers = select_non_overlapping(probabilities > threshold) // C2
//STITCHING STEP
np = number_of_peptides = length(epitope_15mers) / 3 // C3
linker_seq = GGGGS // C4

    for i in 0 to number_of_peptides:

 library_peptide[i] = epitope_15mers[i] + linker_seq +
epitope_15mers[i+ np] + linker_seq +
epitope_15mers[i+ (np*2)] + stop_codon //C5
//COMMENTS

    C1: with random forest model, trained on public epitope dataset (Fig. 2)

    C2: epitope_15mers ordered by probability

    C3: if 2 or less epitopes predicted: no library peptide made,

    for 3-5 epitopes -> one peptide, for 6-8 epitopes -> two peptides, etc

    C4: see Supplementary Fig S2 for evaluation of different linkers
    C5: this puts highest probability 15mer on 1st position of 1st peptide and 2nd highest probability on 1st position of 2nd
peptide etc
    Supplementary Fig. S1 provides an illustration and further explanation.
```

We then used the random forest model to predict epitopes from seven Enterovirus strains and selected 757 peptides with low and high probability. For evaluating antibody binding to these peptides with 55 human samples, we defined a peptide as reactive if at least one of these demonstrated reactivity. The ROC curve (AUC = 0.7) is shown in Fig. 2E, confirming that the model can distinguish epitope-containing peptides from non-reactive peptides based on amino acid composition. For the sake of comparison with an independent epitope prediction tool, we also determined the average BepiPred-3.0 score[15] for the probability of a 15-mer containing an epitope. Depending on the amount of data given to the algorithm as input, the resulting ROC curves have an AUC ranging from 0.54 to 0.74. The lower score is based on the same input data that is used for the random forest; the higher score results from having the surrounding protein sequence inform the prediction. These results indicate that other algorithms may be comparable in performance but would likely involve further adaptation for use in our specific application.

**Dolphyn: a novel algorithm for peptide library design**

Simple tiling methods like Pepsyn[6] divide a protein into peptides of equal length with some overlap (Fig. 3A). This approach has been used to design libraries shown in Fig. 1A, which wastes resources on synthesizing, cloning, and sequencing peptides that have a very low probability to be targeted by antibodies. To improve library efficiency, we developed an algorithm called Dolphyn that selects and combines peptides that have a higher probability of eliciting antibody reactivity.

Dolphyn predicts whether each 15-amino-acid sub-peptide (15-mer) of a protein contains an epitope using the random forest model described above (Fig. 2). For each protein, depending on its length, Dolphyn selects a multiple of three non-overlapping 15-mers with the highest epitope probability (Supplementary Fig. S1). 15-mers are selected if their probability of containing an epitope is greater than 0.5. Then, sets of three 15-mers are combined using Dolphyn's stitching step and separated via a flexible and inert linker sequence GGGGS (see Supplementary Fig. S2 for evaluation of different linkers). A pseudocode description of this algorithm is given in Box 1.

Since the goal of Dolphyn is to represent a smaller, more reactive portion of the original proteome, some epitopes may be missed, especially when there are multiple epitopes located at overlapping positions, or when there are fewer than three epitopes predicted on a protein. Furthermore, because Dolphyn stitches together peptides from different locations across a protein, reactivity data must be interpreted at the protein level.

Dolphyn is available as a modular Python package, with parameters controlling epitope peptide length, the linker sequence, the probability cutoff, the training data for the classifier, and the classifier itself (Fig. 3B). For example, users can replace the classifier with Immune Epitope Database epitopes or their own classifiers.

To evaluate the efficiency of Dolphyn and Pepsyn libraries based on cost-effectiveness, we accumulated the number of reactivities detected per new peptide added to the library (Fig. 3C). In order to

**Table 1 | Peptide library design statistics for Gut Phage Database protein sets**

| | Pilot Library | North American Phageome |
|---|---|---|
| Phages | 112 | 7614 |
| Proteins | 8023 | 588,783 |
| Protein clusters | 3354 | 84,306 |
| Amino acid length of representative proteins concatenated | 750,776 | 15,637,136 |
| Pepsyn tiles | 23,745 | 484,761 |
| Dolphyn tiles | 5266 | 106,762 |
| Pepsyn coverage | 1.77 | 1.74 |
| Dolphyn coverage | 0.32 | 0.31 |
| Compression ratio (Dolphyn/Pepsyn) | 0.22 | 0.22 |

The Pilot Library was synthesized and evaluated in this study.

detect the same number of immune responses as a Dolphyn library, a Pepsyn library requires about three times the number of peptides.

## Compressing the gut phage database for antibody profiling

Camarillo-Guerrero et al.[1] constructed the Gut Phage Database, which contains 142,810 phage genomes derived from metagenomic analysis of stool samples. Roughly one third are marked High Quality and 5% were detected in samples from individuals marked as North American (Fig. 3D). The Gut Phage Database also contains reference amino acid sequences for phage proteins. The authors clustered all proteins at a 95% similarity threshold, resulting in a number of clusters equal to 4% the number of proteins, indicating that the Gut Phage Database contains many homologous proteins.

We used the Gut Phage Database reference to design a phage display library for antibody profiling and evaluating Dolphyn. We considered only high-quality phages that appeared in at least one North American individual, leaving 84,000 protein clusters. A Pepsyn-designed library that tiles these clusters would contain over 480,000 56 amino acid long peptides, whereas a Dolphyn designed library requires only about 100,000 peptides each comprised of three 15-mers separated by G4S linkers (Table 1).

To compare Pepsyn versus Dolphyn library performance, we created a pilot library by selecting the 112 phages detected in 50 or more North Americans. We selected one representative from each protein cluster present in these phages. Using Pepsyn's standard tiling strategy, these proteins are covered 1.77 times using 23,745 56-mers (with 28 amino acid overlaps), whereas Dolphyn covers a third of the proteome, using 5266 56-mers (Table 1). Dolphyn therefore compresses these phage proteomes by nearly five-fold over the traditional approach. In the design of the full gut phage database proteome library, we observe a similar compression.

Figure 3E presents the protein composition of 12 selected phages from the pilot library. Heatmap colors represent the number of peptides in the library for each protein. Many proteins are shared across phages, and phages within Gut Phage Database-defined phage clusters share most proteins. Dolphyn omits some of the smaller proteins where the number of potential epitopes required for efficient sketching is less than three. Consequently, these proteins are not represented in a Dolphyn-designed library, which is one limitation of imposing a required minimum of 3 peptides per protein.

## Effect of stitching on antibody detection sensitivity

Dolphyn creates stitched peptides by combining potential epitopes from the same protein. Specifically, we used the 15-mer with the highest probability determined by the random forest classifier at the first position, followed by the second and third highest probabilities at positions two and three, respectively. In larger proteins that required compression

onto multiple peptides, Dolphyn distributes the highest probability epitopes over the first positions of each stitched peptide so as to maximize the total number of independently reactive peptides per protein.

Our pilot library includes both individual 15-mers and their corresponding stitched versions. Figure 4A shows reactivity data from these peptide sets for four representative samples, where two or more peptides were reactive. We observe that only one individual 15-mer was typically reactive, with a preference for the higher probability epitopes (Fig. 4B). The log fold change values for the stitched version were similar to those of the reactive individual 15-mer (Fig. 4A), indicating effective representation of the epitope. Supplementary Fig. S4 shows all the reactive peptides for one representative sample in the same style as Fig. 4A. Supplementary Fig. S5 indicates how the stitched peptides were validated by the individual epitope library.

## Reactivity of Dolphyn libraries

We profiled plasma samples from 51 healthy individuals using the three pilot sub-libraries. The Dolphyn library contained a three times higher ratio of reactive peptides (log(hfc) > 0 in at least one sample) compared to the Pepsyn library, in which 90% of the peptides were found to be non-reactive (Fig. 4C). Individual predicted epitopes displayed only a slightly higher ratio of reactive peptides compared to Pepsyn. However, it should be noted that these peptides are only 15 amino acids long versus the 56 amino acid long Pepsyn peptides and would therefore be expected to harbor fewer public epitopes if randomly selected.

## Immune response of healthy individuals to gut phages

We then explored the immune response to gut phages in the 51 healthy individuals (Fig. 4D). Using a Phage Aggregate Reactivity score (PhARscore, Methods), we detected antibody reactivity to a cluster (PC_4) of phages in most individuals. Phages in this cluster have a Gut Phage Database-predicted taxon belonging to the *Myoviridae* phylum. The predicted phage hosts in this cluster are primarily *Proteobacteria*, especially *E. coli*.

As the Gut Phage Database does not provide predictions of phage taxonomy or host for all phages, we used BLAST to add annotations to the phage genomes. The Blast *E. coli* heatmap annotation indicates phages with genomes that had an alignment to an *E. coli* reference in the NCBI nt database. These alignments largely correspond to prophage sequences that have integrated into their host bacterium.

## Dolphyn libraries recover observations made with Pepsyn

We confirmed that *E. coli* phages and *Myoviridae*-annotated phages elicited a stronger immune response compared to other phages via Wilcoxon test. The mean PhARscore (Methods) for each phage across all samples is significantly higher in all three annotations for both library designs (Fig. 5A). We noted that the few highly prevalent phages appear to elicit an immune response more commonly across all samples in our cohort, which includes only North American individuals (Fig. 5B).

We investigated what protein level targets drive the immune response to the highly reactive phage cluster in healthy individuals. Figure 5C displays the number of reactive peptides in the proteins of the four most reactive phages in five individuals with the most robust antibody responses according to their PhARscores. We observe that some proteins are detected by several individuals' antibodies, but overall, the individuals exhibited distinct immune response profiles. The proteins are ordered by size, and a higher number of reactive peptides is expected and observed for larger proteins (towards the left).

Dolphyn-designed libraries demonstrate similar discovery power (accuracy) for identifying protein antibody targets in 53 individuals, as shown in Fig. 5D. Dolphyn peptides only recall about a third of the proteins (131) that are reactive in Pepsyn (469) partially because Dolphyn does not include some proteins. However, Dolphyn detected several protein targets (79) that were not reactive in the Pepsyn library, potentially due to multiple independent epitopes being

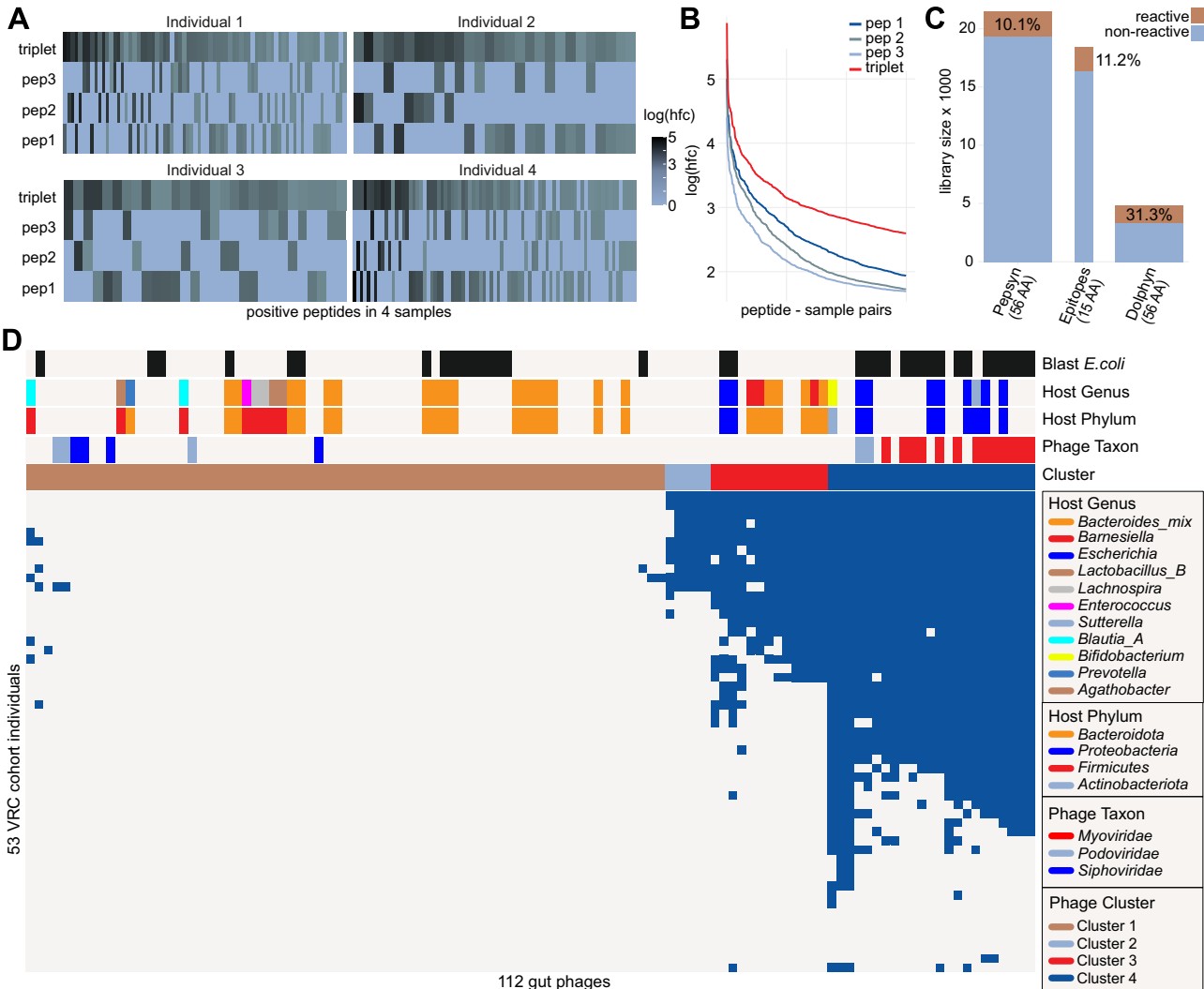

**Fig. 4 | Healthy individuals' antibody reactivity to 112 gut phages in pilot library. A** Relationship between reactivities of stitched Dolphyn peptides and unstitched predicted epitope peptides. For four samples, all peptide quartets (three 15-mers and their stitched version) are shown where two or more peptides show reactivity. Peptide sets are ordered by highest to lowest reactivity from left to right. **B** The 15-mer peptides were stitched in the order of their probability score on the combined peptide, starting with the highest at position one. Accordingly, a difference in mean log reactivity score can be observed. Peptides are ordered by reactivity score per sample on the x-axis. **C** The phage proteomes in the pilot library were represented using three different approaches: Pepsyn, predicted epitopes,

and Dolphyn. The Pepsyn library with regularly spaced 56-mer probes and is the largest but least reactive per peptide. The predicted epitopes alone achieve a similar proportion of reactive peptides despite the probes being small (15-mer amino acids (AA), depicted by width of bar). The Dolphyn library achieves a greater proportion of reactive probes than the alternatives. **D** Anti-phage antibody reactivities in healthy individuals of the Vaccine Research Center (VRC) cohort, depicted as binarized PhARscores, a phage-level aggregate reactivity score. The annotation on the top shows predicted phage properties and phage clusters. | Source data are provided as a Source Data file.

simultaneously bound by a combination of weak antibody interactions. When including these reactive proteins as ground truth, Dolphyn achieves higher performance (Supplementary Fig. S3).

The pilot phage library designed with Pepsyn produced several key observations that were recapitulated by the Dolphyn library. The Principal Component Analysis shown in Fig. 5E is based on sample PhARscores. Color is used in the plot to highlight phage attributes. The rightmost plots show the four identified phage clusters of the heatmap, initially found using the Pepsyn-designed library (Fig. 4D). We observed that the Dolphyn-designed library preserved the same clustering for our cohort.

## Discussion

Here we introduce the Dolphyn algorithm for efficiently converting large antigenic spaces into tractable peptide libraries for antibody profiling applications. The algorithm reduces the number of peptides

in a library for a given proteome by 78%. Conversely, Dolphyn peptides are roughly three times as likely to be reactive, versus peptides from uniformly tiled proteins. Dolphyn peptides recall 29% of the gut phage proteins that exhibited reactivity with Pepsyn designed peptide tiles. Some of the lost recall is attributed to Dolphyn's exclusion of specific proteins from its representation (e.g. proteins with fewer than three predicted epitopes).

Dolphyn employs a random forest model to select peptides predicted to contain an epitope based on their amino acid content. Training the model on public epitopes, we discovered that the majority of public linear epitopes span about 15 amino acids and that side-chain information appears to be the most influential factor for distinguishing peptides with and without epitopes. We apply this principle to compression of synthetic antigenic spaces. However, our epitope prediction model could also be used in vaccine research or other immunobiology applications. In the future, it will be important to

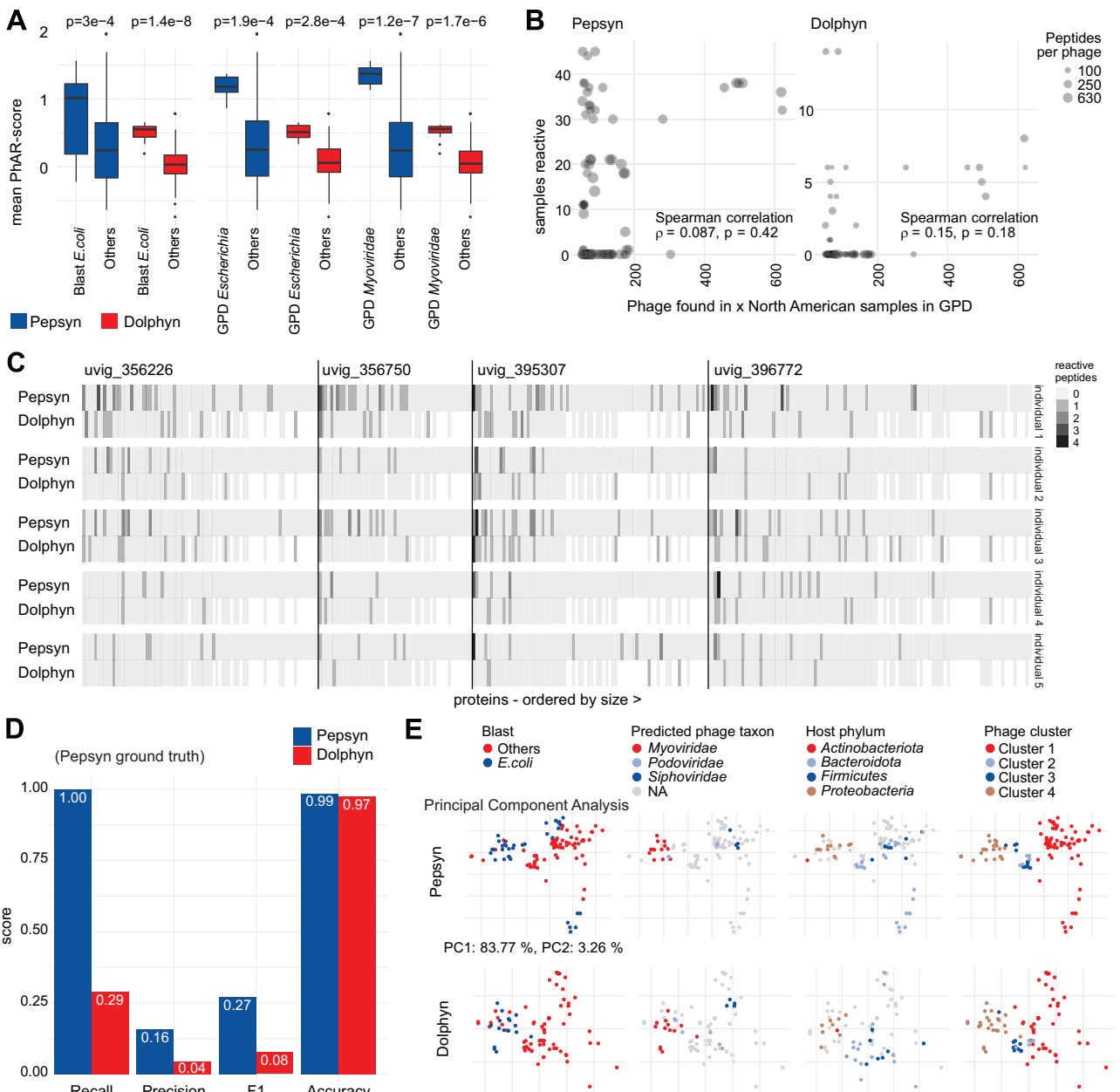

**Fig. 5 | Comparison of Dolphyn and Pepsyn results. A** A significant difference in phage reactivity can be observed in mean PhARscore of Pepsyn and Dolphyn designed libraries, when splitting the 112 (Pepsyn) and 88 (Dolphyn) phages into two groups according to their taxonomic annotation (2-sided Wilcoxon test). The horizontal lines in the boxplots indicate the median, and the lower and upper value of the box the interquartile ranges (25 and 75 percentiles). The whiskers extend to the most extreme data point which is no more than 1.5 times the interquartile range from the box. **B** The phage reactivities detected in more samples in the Gut Phage Database (x-axis) were also reactive in more individuals in our cohort (y-axis). Phages were only considered if present in both library designs. The spearman correlation is calculated based on reactivity and *p*-values reported based on a 2-sided test. **C** Phage proteins from phages with PhARscores indicating high

reactivity are displayed in sequential order (no shared proteins) for 5 individuals. Note that the Dolphyn library contains less peptides per protein and has consequently a lower number of reactive peptides per protein. **D** Protein discovery power. Using reactivities to Pepsyn proteins as ground truth, performance metrics are shown for both Pepsyn and Dolphyn peptides in recovering protein by sample reactivities. **E** Principal component analysis clustering of phages colored by different annotations, with the first principal component (PC) on the x-axis and the second PC on the y-axis. The PCs of phages according to the sample-reactivity vectors per phage are similar for both libraries. Phages with same annotations and clustering in the heatmap (Fig. 4) group together independently of the library design. | Source data are provided as a Source Data file.

investigate whether using outputs from more complex algorithms like Alphafold2[21] can improve predictions. The Dolphyn pipeline is set up to accept new models.

The algorithm's stitching step combines three potential epitopes into one library peptide. While the prediction model enables the prioritization of protein regions during compression, the stitching step notably enhances peptide reactivity.

Our algorithm is aimed at maximizing detection efficiency of immune responses to very large protein sets. We have focused in this study on identifying features of linear public epitopes, which would require the fewest number of peptides to detect the greatest number of individual immune responses. Notably, including private epitopes would by definition detect only very few individual responders per library member. It is important to note that features

predictive of private epitope selection may be similar or entirely different.

Additionally, it should be acknowledged that anti-T7 phage antibodies, if present, might introduce background noise to assay results. To enhance the sensitivity of future assays, we recommend considering the incorporation of a PCR spike-in control, which could facilitate the detection and quantification of this library-wide background noise.

The trade-off for a compressed library design is reduced antibody binding resolution. Utilizing Dolphyn libraries, we explored general antibody reactivity to proteins or species rather than identifying specific epitopes. In case an investigator prefers to maximize sensitivity and mappability at the expense of encoding efficiency, an enhanced tiling approach can be employed, as recently demonstrated by Na[22], which targets small and tractable antigenic spaces, for example a reduced set of viral species.

Using phage display libraries designed with Pepsyn and Dolphyn, we studied the immune response of healthy individuals to gut phage. We found that the majority of individuals showed reactivity to *E. coli*-infecting *Myoviridae*. Both libraries captured this relationship. It remains to be determined whether these antibodies are functional (e.g. enhancing or neutralizing), and what sorts of health-related phenotypes, if any, associate with these immune responses.

In short, Dolphyn libraries require much fewer peptides to reveal key patterns in antibody reactivity, giving it an advantage over libraries that contain regularly spaced overlapping peptides. The modularity of Dolphyn, such as the interchangeability of the epitope prediction module, further highlights its potential for future applications in peptide library design for the immunological study of large proteomes, such as the entire gut, skin, or lung microbiome.

## Methods

We confirm that our research complies with all relevant ethical regulations. We used data and serum samples from previously established cohorts and studies that were approved by the Vaccine Research Center (VRC)/National Institutes of Allergy and Infectious Diseases, the Institutional Review Board of Johns Hopkins University and The Colorado Multiple Institutional Review Board. All samples were obtained under informed consent.

### Peptide libraries and cohorts

**Public epitope library (for the study of the nature of epitopes and training the random forest classifier).** This T7 bacteriophage display library contains 357 56 amino acid long *wildtype* peptides that showed frequent antibody reactivity (public epitope peptides) in a previous VirScan study[7]. Within each of these wildtype peptides, a series of shorter peptides of length 15, 20, 25, 30, 35, 40 and 45 amino acids were designed to tile across the original 56 amino acid peptide in steps of 5 amino acids (Fig. 1B). In addition, each of the wildtype peptides was subject to triple alanine mutation scanning, including peptides identical to each of the native peptide sequence but with triple alanine substitutions scanning from N- to C-terminus of each peptide. This library contains 46,070 peptides, and has also been used by Shrock et al.[14] with a different cohort. The amino acid sequences are available as row names within the Public Epitope Data Set file hfc_pubEpitopes.csv on Zenodo.

**Gut phage database - phageome pilot library (for evaluating the performance and demonstrating the utility of the Dolphyn algorithm).** This library contains 48,128 peptides that are 56 amino acids long and is divided in three subsets of peptides, representing the same 112 prevalent phages in 3354 protein cluster representatives:

1) 19,117 peptides (length = 15 amino acids) that are likely to contain an epitope based on the random forest predictions (value > 0.5). These encoding oligonucleotides are padded on the 5' end to make them the same length as the other two peptide libraries (56 amino

acids), with three stop codons and a random sequence generated with a pseudo-random generator, i.e. the Python random.choice() function.

2) 5266 peptides designed with the Dolphyn algorithm. 15-mer epitope peptides are grouped if they are present together on more than one protein. A *Dolphyn peptide* is created for every three epitope 15-mers, that are available per protein group. The 15-mer having the highest-probability epitope goes first, then a GGGGS linker, then the 15-mer having the second highest probability, then a GGGGS linker, then the 15-mer having the third highest probability, then a stop codon, creating a peptide of 56 amino acid length. If two or more Dolphyn peptides are created per protein set, the second highest probability 15-mer gets the first position on the second peptide and all other epitopes are ranked and positioned accordingly.

3) 23,745 Pepsyn peptides created by tiling the protein sequence with 56 amino acid long peptides and overlapping by 28 amino acids. If the protein length is not a multiple of 28, a full 56-mer is created at the C-terminus of the protein, potentially overlapping more than 28 amino acids of the previous tile.

All three sub-libraries were reverse translated with the Python Pepsyn package's revtrans command to obtain 168 nucleotides long oligonucleotides. A 16 nucleotides long prefix (AGGAATTCCGCTG CGT) and suffix (ATGGTCACAGCTGTGC) were added to each oligo for PCR amplification, making the oligonucleotides 200-nt long. Sequences of oligonucleotides for all three sub-libraries are available in the file PhageScan_PeptideAnno.tsv on GitHub. The oligonucleotide library was synthesized by Twist Bioscience (San Francisco, CA).

**Enterovirus sublibrary (for evaluating the prediction performance of the random forest classifier).** This library contains 1514 peptides derived from seven reference enterovirus sequences. The sequences were selected using the cd-hit tool to represent each species of Enterovirus A, B, C, D and Rhinovirus A, B, C. Based on the selected sequences, we designed two sets of peptides using Dolphyn and Pepsyn algorithms. A first set of peptides consists of 757 epitope peptides with a length of 15 amino acids were selected using the random forest scoring method. An identical second set was generated by adding a stop codon to the C-terminus of the first set of epitope peptides.

To convert the designed amino acid peptides to oligonucleotide sequences of uniform length, the Pepsyn algorithm was employed. The function 'revtrans' reverse-translated the amino acids sequences into DNA sequences by randomly choosing codons based on the *E.coli* codon usage table, with a frequency threshold of 0.1. Since the designed peptides varied in length, they were padded to a length of 120nt with a linker sequence, GCAAGTCCTGCAGCTCCAGCCCCT GCAAGCCCAGCAGCTCCAGCACCAAGTGCACCTGCTGGCGGAGGAG GTTCTGGCGGGGGCGGGAGC. Prefix AGGAATTCCGCTGCGT and suffix GTCGTGACTGGGAAAC were added for cloning purposes. Pepsyn's 'recodesite' command was used to eliminate all EcoRI (GAATTC) and HindIII (AAGCTT) sites in the oligonucleotide sequences, as they were used to clone the library inserts into the T7 vector. Sequences of oligonucleotides for all three sub-libraries are available in the file EnteroEpitopes.csv on GitHub. The oligonucleotide library was synthesized by GenScript Biotech (Piscataway, NJ) using their oligonucleotide library synthesis platform.

**Enterovirus (EV) library screening cohort (used with the Enterovirus sublibrary for evaluating random forest prediction performance).** The Diabetes Autoimmunity Study in the Young (DAISY) (clinicaltrials.gov identifier: NCT03205865) is a longitudinal study of children at high risk for development of Type 1 Diabetes (T1D) due to genetic markers or family history. The cohort comprises approximately 7% African American, 30% Hispanic, and 63% non-Hispanic white, with the remaining participants being of biracial or other ethnicity. The study follows participants from birth, collecting blood samples annually for

autoantibody testing and other biological samples for future analysis. If T1D-related autoantibodies are found, the subjects are closely monitored for the onset of the disease. The DAISY cohort was recruited between 1993 and 2004, and follow-up data is available until February 2018[23]. To conduct a PhIP-Seq screen with the Enterovirus sublibrary, a total of 55 patient samples were selected from a subset of six subjects, consisting of three T1D positive and three T1D negative individuals, three male and three female, aging 0-7 years. Other meta-data, such as sex/gender and age, were not provided to the study team.

**Vaccine Research Center (VRC) cohort (used with the Gut Phage Database Phageome Pilot Library, and previously with VirScan (EBV results)).** The Vaccine Research Center (VRC) cohort has been described previously[24] and is comprised of 801 healthy community volunteers in the greater Baltimore/Washington DC area recruited for research studies. Of the 801 individuals, 535 are of European genetic ancestry, 194 of African genetic ancestry, 32 of Asian genetic ancestry and 40 belonging to other ancestral groups. The VRC cohort included 446 men, 351 women and 4 unknown. Their ages ranged from 18 to 70 years, with an average of 35.79 years. VirScan was performed on all 801 VRC subjects. A subset of 50 subjects were selected for a PhIP-Seq screen with the Gut Phage Database pilot library set. Of the 50 individuals, 41 are of European genetic ancestry, 4 of African genetic ancestry, 4 of Asian genetic ancestry and 1 belonging to other ancestral groups. The group consisted of 22 men and 28 women between ages 18 and 64, with an average age of 39. The study was approved under the NIAID/NIH protocol "VRC000: Screening Subjects for HIV Vaccine Research Studies" (NCT00031304).

**Public Epitope Data Set cohort (used with the Public Epitope Library to create the Public Epitope Data Set).** 425 plasma samples (Supplementary Table S1) for public epitope testing were obtained from the Genital Shedding (GS) Study (Uganda and Zimbabwe; 2001–2009) in 59 women, which evaluated the relationship between hormonal contraceptive use, genital shedding of HIV, and HIV disease progression among individuals with known dates of HIV seroconversion, aging 19–37 years[25]. Antiretroviral treatment was recommended for study participants with CD4 cell counts below 250 cells/mm$^3$, consistent with local treatment guidelines at the time the GS Study was performed. Data for CD4 cell count and viral load were collected in the GS Study[25] data on the timing of antiretroviral treatment initiation was obtained by review of clinic records. For the Public Epitope Data Set, samples were treated independently, as individual antibody responses change over time.

**Statistics & reproducibility.** We worked with data and serum samples from previously published cohorts and studies, for which no statistical method was used to predetermine sample size. No data were excluded from the analyses, except when stated explicitly for a sub-result. The exclusions are traceable in the published analyses scripts. Randomly selected subsets of samples were used from the VRC cohort and the EV cohort to generate certain datasets. The investigators were blinded during sample selection and outcome assessment.

## Experimental methods

**Library construction.** Each oligonucleotide pool was resuspended in ultrapure H$_2$O to a concentration of 10 ng/μl. A first round of 2 cycles of PCR was performed using 1 ng of library DNA and the primers GCGCAAATGGGCGGTAGGCGTGAGGAATTCCGCTGCGT (forward) and GATTAACCCTCACTAAAGGGAAAGCTTGCACAGCTGTGACCAT (reverse). The PCR product was purified and a second round of 12 cycles of PCR was performed on all recovered PCR product with the primers CGCAAATGGGCGGTAGGCGTG (forward) and ATTAACCCTC ACTAAAGGGA (reverse). The amplified DNA was purified using a PCR

purification column, digested with EcoRI and HindIII, gel purified and ligated with EcoRI/HindIII digested T7FNS2 vector arms according to the manufacturer's instructions[26]. The ligated inserts were packaged with the T7Select packaging kit (Millipore Sigma, St. Louis, MO) as per manufacturer's instructions. An adequate number of packaging reactions to ensure a 100X coverage of the library were set up, pooled and a pre-amplification phage stock was prepared by the plate amplification method. The pre-amplification phage library stock was titered, mixed with DMSO at a final concentration of 10% and stored at –80 °C for the long term. A post-amplification working stock of the library was prepared for PhIP-Seq, using the liquid amplification method, ensuring that a minimum plaque forming units (pfu) of at least a 100X the library size was used as input. The post-amplification library was titered and a pfu of 100,000X the size of the library was mixed with each sample for PhIP-Seq.

**Phage immunoprecipitation sequencing (PhIP-Seq).** PhIP-Seq was performed as previously described[12]. Briefly, 0.2 μl of serum sample was incubated with the phage library overnight at 4 °C. The serum-phage mixture was then incubated with a mixture of 20 μl magnetic protein A beads (Invitrogen cat# 10001D) and 20 μl protein G beads (Invitrogen cat# 10004D) for 4 h to immunocapture serum IgG antibodies and antibody-bound phage. The phage-antibody complexes captured on the beads were washed to remove unbound phage. After bead washing, peptide-coding DNA inserts from the phage were PCR amplified with forward and reverse primers containing dual indexed adapters suitable for Illumina sequencing. The PCR amplicons were pooled and subjected to DNA sequencing on an Illumina NextSeq 500 instrument.

## Computational methods

**Preliminary analysis of PhIP-Seq data.** The output from short-read sequencing of the immunoprecipitated phage libraries underwent initial processing using the edgeR pipeline as previously described[27]. Particularly, sequencing reads of a phage library member were counted via exact matching of the first 50 peptide coding nucleotides and a pseudocount was added. The magnitude of reactivity to each library member relative to mock immunoprecipitations, as defined by a fold change and associated p-value, was determined using the edgeR package in R[28]. Significant reactivity or hits were defined as library members with counts greater than 15, fold change greater than five, and p value less than 0.001 (referred to as hit foldchange (hfc) throughout this manuscript). All other analysis of PhIP-Seq data was performed subsequent to this initial processing.

**Phage aggregate reactivity score (PhARscore).** To facilitate interpretation of complex antibody reactivity profiles, we modified the ARscore algorithm to aggregate antibody reactivity to all peptides that represent each phage[29]. The Gut Phage Database contains many homologous proteins, resulting in peptides that represent multiple proteins and phages. Phage-association of each peptide was tracked at every clustering step during the library design. Phage aggregate reactivity scores (PhARscores) were calculated for each phage represented by ≥25 peptides in a sublibrary (112 phages in the Pepsyn sublibrary, 88 phages in the dolphyn sublibrary). PhARscores from each sublibrary were generated separately.

PhARscores for a given phage were calculated by comparing mean log2 foldchange of each phage-associated peptide set to distributions of mean log2 foldchange values of the same number of randomly drawn peptides from the same sublibrary. This process was then repeated whereby, in each iteration peptides from strongly reactive phage (PhARscore >1) were removed from the pool of peptides used to generate random distributions. This process was performed a maximum of seven times or until no new phage met the reactivity threshold.

**ML training set.** The 15-mers of the Public Epitope Data Set serve as training data. A 15-mer from the Public Epitope Data Set is considered to be reactive if 2 or more samples have a log(hit-foldchange) >0. For negative examples, we choose 15-mers with high count on the (empty) bead samples with no reactivity in any sample, to avoid including sequences that may not show reactivity due to technical reasons. The dataset was constructed to have the same number of positive and negative examples (balanced dataset). The advantage of this dataset is that many similar training examples (sequences) are contained with differing labels. When splitting test- and training-set (5/95%), we ensured that sequences derived from the same wildtype were not present in both sets.

**Random forest classifier.** For binary classification of 15 amino acid long peptides, a Random Forest model (Python scikit-learn package version 0.24.2 *RandomForestClassifier*) was trained on 556 features (Fig. 2D). Default values for model parameters were used, including the number of trees ($n_{estimators} = 100$) and setting the random_state = 42 for reproducibility.

A random forest model combines several features' impact. A particular feature's importance (impurity based) is measured and can be extracted from the model. We report the top 9 features in Fig. 2C. A higher importance value indicates that the feature is more effective at distinguishing the two classes. The *RandomForestClassifier* also provides out-of-bag scores, summarizing the prediction performance of the random forest model on out-of-bag samples, which were used for Fig. 2A.

**EBV random forest testing.** 2263 Epstein-Barr Virus (EBV) peptides, 945 Rhinovirus peptides and 4016 Human cytomegalovirus (CMV) (independent 56-mers, Pepsyn design) from the VirScan library were used to assess our model predictions. 801 samples from the VRC cohort were used to establish a ground truth as to whether a peptide is reactive or not. A positive label (to contain a public epitope) is given when at least eight members (1%) of the cohort showed reactivity. All sub-15-mers were evaluated with the Random Forest classifier. The mean probability of all 15-mers in the 56-mer determines the probability score for the peptide, from which the ROC curve in Fig. 2B was constructed.

**BepiPred epitope prediction on enterovirus sublibrary peptides.** For the ROC curve generated in Fig. 2E, BepePred-3.0 was run with default settings through https://biolib.com/DTU/BepiPred-3 in two experiments. The first included the 15 amino acid long sequences of the Enterovirus sublibrary. The second experiment used the seven reference enterovirus sequences as input. In both experiments, the BepiPred scores for an epitope at each residue were averaged over the 15 amino acids contributing to a peptide. For creating the ROC curve, the same labels were used as in the validation of the random forest, derived from the Enterovirus library screening cohort.

**Principal component analysis.** The Principal Component Analysis (Fig. 5E) was conducted with the base R (version 3.6.3) function prcomp (*stats* package) based on the PhARscore vector of all samples for each phage. The first two principal components were plotted and per panel colored differently for various phage meta information.

**BLAST for additional phage annotation.** To annotate phage genomes according to whether they potentially infect *E.coli* or other Bacteria, we used blastn version 2.13.0+ to scan the entire NCBI nt database for similarity. We considered that if a phage genome was contained in a bacterial genome in the database, the phage may have infected that bacterium and was sequenced alongside when the reference genome was created. A BLAST hit to a viral taxonomy might indicate a potential taxonomic annotation for these novel phages.

For the binary annotation in this manuscript, an assignment is used that indicates whether an *E.coli* genome is among the top 15 BLAST hits that have an *e*-value < 1E−6.

**Metadata analysis of the gut phage database.** The Gut Phage Database[1] contains metadata for all phages. The predicted phage taxon and predicted host was used to annotate the heatmap in Fig. 4D. Furthermore, this metadata contains a list of sample identifiers, corresponding to the cohort used in the study, of whose metagenomic samples the phage genomes were derived. We counted the amount of samples annotated as North American to i) select the locally prevalent phages in our pilot library and ii) conduct the prevalence study in Fig. 5B.

## Reporting summary
Further information on research design is available in the Nature Portfolio Reporting Summary linked to this article.

## Data availability
All relevant data supporting the key findings of this study are available within the article and its Supplementary Information files. The processed PhIP-Seq and public epitope reactivity data are available on Zenodo at https://doi.org/10.5281/zenodo.7979556. The further processed data used for figures and tables are provided in the Source Data file. Source data are provided with this paper.

## Code availability
The GitHub repository contains scripts, such as the Jupyter Notebook scripts for deriving the results (folder Manuscript Analyses), the Dolphyn python package and the machine learning models as described in the manuscript. It is available at https://github.com/kepsi/Dolphyn and on Zenodo (https://doi.org/10.5281/zenodo.7979556)[30].

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

## Acknowledgements

We thank the members of the NIH Vaccine Research Center for healthy volunteer serum sample collection: M. Roederer, B. Graham, L. Novick, J. Casazza, J. Ledgerwood, U. Sarwar, L. Chang, C. Starr Hendel, L. Holman, S. Plummer, P. Costner, I. Gorden, B. Larkin, F. Mendoza, J. Saudners, K. Zephir, M. E. Enama, G. Yamshchikov, I. Pittman and P. Williams.

This work was made possible by National Institute of General Medical Sciences (NIGMS) grants GM136724 (IR and HBL) and R35GM139602 (BL), a grant from The Leona M. and Harry B. Helmsley Charitable Trust (RSL, JLR, and HBL). The DAISY cohort has been funded by the NIH DK032493-37 (MR and KW). CM has been funded through Family Health International (FHI) (N01-HD-0-3310). SE is an investigator with the Howard Hughes Medical Institute.

## Author contributions

This project was conceived by HBL, IR, BL and AML. The public epitope library was previously designed by HBL, TK and SE, and the associated Genital Shedding cohort established by CM. AML designed the Phageome libraries, including validation libraries (Pepsyn, single epitopes, triplets, public epitope validation and Enterovirus tiling). The DAISY cohort was provided by KW and MR. The Enterovirus validation library was constructed by MN. TV oversaw experimental analysis of the Phageome libraries using the VRC cohort. WRM developed and derived the PhARscore metric. AML developed the prediction model and Dolphyn algorithm and performed all analyses with the support and guidance of BL, HBL and IR. The statistical approaches were advised by IR. Figures were prepared by AML. All authors read and approved the final manuscript. RL and JR helped with the interpretation of results, operational considerations and funding the project.

## Competing interests

HBL and SE are inventors on an issued patent (US20160320406A) filed by Brigham and Women's Hospital that covers the use of PhIP-Seq for antiviral antibody detection; HBL is a founder of Portal Bioscience, Alchemab, and Infinity Bio. TK is a founder of TSCAN Therapeutics. SE is a founder of TSCAN Therapeutics, MAZE Therapeutics and Mirimus, serves on the scientific advisory boards of Homology Medicines, MAZE Therapeutics and TSCAN Therapeutics, and is an advisor for MPM Capital, none of which affect this work. The remaining authors declare no competing interests.
