## [Peer Review File · Nature Communications]

Efficient encoding of large antigenic spaces by epitope prioritization with DolphynReviewer #1 (Remarks to the Author):

In this study, Liebhoff et al develop an algorithm to improve the efficiency with which large epitope spaces (and specifically, the gut phageome) can be represented in high-dimensional peptide-based serological assays. They use a large experimentally-generated training data set to build a model for linear epitope prediction, and then use this model to generate a library with which they profile anti-phage antibodies in healthy subjects. This is an important area of research given the fact that phage represent a highly-diverse and under-explored space, and the immune response to them has the potential to be a powerful modulator and reporter of the state of the microbiome. However, enthusiasm for the manuscript is reduced by the following concerns that go to the level of innovation, impact and validation of the findings.

1. The authors do not establish the innovation of their RF classifier over the various published epitope prediction approaches. A number of algorithms for the prediction of B cell epitopes (include linear epitopes) from primary amino acid sequence have already been described (e.g. PMIDs: 36366745, 23667458), using increasingly sophisticated machine learning approaches. These existing models are trained on datasets of similar size or larger than the one presented here, and show comparable/superior performance (AUC \sim 0.6-0.8). The authors do not cite any of these studies, nor attempt to put their findings into that context (eg by comparing the performance their model side-by-side with existing ones).

2. Overall, Dolphyn likely represents only an incremental step towards the authors' stated goal of addressing the vastness of the gut phageome. The authors show that Dolphyn achieves a \sim 3-fold enrichment of reactive peptides over Pepsyn (Fig 4C): however the majority of this effect (\sim 2X) would be expected from the simple step of eliminating the 2X tiling redundancy of Pepsyn. (It is also important to acknowledge that this 2X compression removes the internal replication that was an intentional part of the original VirScan design approach). Overall, therefore, the impact of the RF classifier therefore appears to be modest, which is consistent with the observed AUCs.

Relatedly, the following conclusion in the abstract "Dolphyn improves the fraction of gut phage library peptides bound by antibodies from 10% to 31% in healthy individuals, while also reducing the number of synthesized peptides by 78%", seems like a case of double-counting. It would be more appropriate to describe these effects disjunctively: a \sim 3X increase in the proportion of reactive peptides, OR a \sim 3X reduction in the number of peptides that need to be synthesized to achieve a fixed number of reactive peptides.

3. The profiling of antibody responses to gut phage in healthy subjects (Figs 4,5) is intriguing. The interpretation of these findings, however, is difficult in the absence of any attempt to validate these findings against expected reactivities / orthogonal measures / covariates.

Reviewer #2 (Remarks to the Author):

This manuscript describes the development and application of a new approach and package for designing PhiP-Seq libraries. This package called Dolphyn reduces the number of peptides needed in a library by removing those that don't have amino acid features of epitopes, and then 'stitching together' predicted 15-mer epitopes from the same protein with a GGGGS linker to generate a 56-mer containing 3 15-mers. Using this approach the authors show that at a phage library level the Dolphyn approach reduces library size markedly, and predicts similar peptides compared with the Pepsyn tiling approach. The paper is clear and the rationale and approach is well described.

My major question is around interpretation of findings/epitopes using the new Dolphyn approach - once a 56mer is identified as a possible hit, what should be done next to validate this as a biomarker/public epitope? The 56-mer contains 3x15mer epitopes but the current paper does not show any post-PhiP-seq analysis or validation to support their selection of the 'most likely' epitope as the 1st 15-mer, followed by the 2nd most likely and then the 3rd. Is this likely to be confirmed or reflected in peptide ELISA or similar using the 3x 15-mers? Or is it suggested that the stitched peptide is used for orthogonal validation? Have the authors performed any follow on analysis along

these lines from the hits they have identified to show the value of DolphyIn beyond Phip-Seq?

Minor comment - this paper is quite heavy on acronyms and package names. Some could be spelt out in full, or referred to in full at first use, especially for non-bioinformatic audiences eg DIAMONDS, GPD, VRC, PEDS etc

Reviewer #3 (Remarks to the Author):

Summary

Liebhoff et al. report interesting work that can definitely deserve publication in Nat comms. They have generated relevant new insights not only for researchers using PhIP-Seq, but also more generally into immunology, the nature of antibody bound epitopes etc. The Dolphyn model generated is probably also relevant for researchers designing peptide arrays with short x-mers. Regarding the presentation, the paper covers two "stories" at once: 1. A new method to reduce the epitope space for antigen library design (called Dolyphn) and 2. studies on anti-gut phage antibody responses. The 1st part feels stronger and is the main focus, the 2nd part lacks key insights into the biology (if there are Igs against the phages, do the abundances of the respective host bacteria change etc.). However, as the title is focused on the 1st part, I do not see an issue with that. The phage antigens can be seen as validation example for the algorithm, and for that they serve their purpose.

The experimental work is extensive, the controls are sound, application of ML nicely carried out with held out controls sets. I see little issues with the experimental work, and my comments below relate mostly to the presentation and interpretation. Since there are quite a few comments, I have recommended major revision, but this does not affect my assessment that the paper is well suited for Nat comms.

Major comments

- Some of the more interesting insights in the Dolphn design are a bit buried in the manuscript (e.g. a key point "This suggests that we can distinguish peptide sequences more likely to contain epitopes by their amino acid composition, which defines the primary and partially secondary structure of proteins."). This is quite a fundamental insights, which amino acid features are predictive of antibody binding – these findings are not highlighted. The authors may want to highlight this in a more pronounced way.
- The beginning of the Results section details two approaches of profiling the PEDS dataset, 1.) alanine scan and 2.) the k-mer scan. While the alanine scan approach is decently explained, I did not find a clear explanation of the k-mer approach and why it was performed – please add in the main text.
- "To assess performance on an independent dataset, we collected a set of Epstein-Barr-Virus (EBV) peptides that had been previously screened in a study by Monaco et al. (2021)."
 - o Authors Larman and Elledge have published a lot more datasets. Why not run Dolphyn on more of those as proof of concept? Why exactly on the EBV data? Couldn't you take the full library content of any other paper, take the peptide sequences as input for Dolyphn and compare the prediction outcome to the actual antibody binding signal previously measured? That would strengthen the paper.
- "This suggests that we can distinguish peptide sequences more likely to contain epitopes by their amino acid composition, which defines the primary and partially secondary structure of proteins."
 - o What about non GRAB motives, in the Shrock 2023 Science the authors state: "This work has several implications: first, it suggests that private rather than public neutralizing antibodies may be superior candidates for inclusion in therapeutic monoclonal antibody cocktails, because private antibodies are less likely to exert population-wide selective pressures on pathogens and may thus retain efficacy for future variants."
 - ♣ According to that, wouldn't some private epitopes be missed by the training on shared epitopes?
 - ♣ This is a valid way to reduce the epitope space and not against this manuscript here, just mention these considerations to let readers judge if they want to use this or not.
- The whole paper is not written in the most clear way, and in parts a bit confusing, (such as the bit on k-mers and ala scan mentioned above), if possible streamline a bit, maybe add an overarching figure/schematic that connects everything and add a paragraph in the beginning of the

results were you briefly outline what you are doing. I was reading this and not knowing what to expect or why certain things were mentioned at first, later most made sense. Having an outline of what is about to come would have made it easier to read the paper.

- "Simple tiling methods like Pepsyn (Larman et al. 2011) divide a protein into peptides of equal length with some overlap (Fig 3A). This approach wastes resources on synthesizing, cloning, and sequencing peptides that are not reactive."

- o Not sure I agree, related to above comment. Would antibody bound GRAB and none. GRAB motives be equally well predicted? The notion of the statement above (if I understand the process correctly applies mostly to GRAB motifs, what is the overall ratio of GRAB vs. non-GRAB antibody bound motifs? Discuss briefly.

- The paper focuses heavily on the benefits of Dolphyn, but does not highlight the limitations:

- o "In order to detect the same number of immune responses as a Dolphyn library, a Pepsyn library requires about three times the number of peptides."

- ♣ It needs to be mentioned how many signals would be missed (false negatives in a way) using the Dolphyn approach.

- o In other words, in "Reactivity of Dolphyn libraries" the absolute number of bound peptides need to be mentioned, how many more were only detected with regular tiling, than with the prediction and stitching approach?

- o I think this comes up later in a way "Dolphyn peptides only recall a bit more than half the proteins that are reactive in Pepsyn partially because Dolphyn does not include some proteins."

- o But should be mentioned first in the relevant section where readers would look for this info.

- o "The algorithm reduces the number of peptides in a library for a given proteome by 78% and triples the reactivity per peptide, as compared to uniform protein tiling"

- ♣ This should followed with a sentence like "Compared to the reactivities in the x-fold more extensive/variant intensive conventional tiling, Dolphyn misses approx. half the proteins of the conventional tiling, because they are simply not included in the Dolphyn library." (you can sure write this in a better way)

- ♣ Here it should also be highlighted, that there is hence a tradeoff between false negatives (missed ones, because they were simply not included) using this approach – if one is interested in a particular species or set of antigens, it may be better to represent them by full tiling, rather than Dolphyn.

- Key limitation to mentioned: with 50 to 90 aa peptides at least some secondary structures and semi-conformational epitopes would be represented, with 15 mers only linear epitopes.

Minor comments

- How large is their library, how large cohort?

- What about reactivity against T7 phages? Those antigens could not be included as part of the library, on the other hand there may be some antibody responses against T7 phages, that would react against the phage library independently of the antigen displayed on a phage's surface. That would potentially increase background noise and result in low level binding against the entire library – discuss implications/mention this issue.

- P. 3 "a cohort of 59 individuals (425 samples)" – what done with multiple samples per person?

- Anti-gut phage antibody responses have been measured with PhIP-Seq in these papers, discuss briefly (the info on the phages is there somewhat scarce, but they show up in the associations etc. reported): PMIDs 37164013 and 37164015

- Public Epitope Data Set (PEDS) – is this available as supporting table/xlsx/csv file? Should be added or explain why not added.

- Methods:

- o Public Epitope Library – this is not linked to the Shrock 2023 Science paper? Because no reference to that paper is given.

- o "The final Dolphyn library is comprised of 48,128 individual 200-mers."

- ♣ Maybe start out with that first, and then explain that sub libraries.

- o Enterovirus Sublibrary: What's the total number of variants here?

- o Did you observe any differences in the quality / error rates / uniformity between the library synthesized by Twist Bioscience (San Francisco, CA) and the other one by GenScript Biotech (Piscataway, NJ)? Could there be any bias affecting the results?

- Author list order in the manuscript and online system is different – on purpose?

- "For each protein, Dolphyn selects the three non-overlapping 15-mers with the highest epitope

probability (Fig S1). For long proteins, multiple sets of three 15-mers are selected if the probability of containing an epitope is greater than 0.5."

o Discuss why this approach was chosen. What is a "long" protein? Say you take the SARS-CoV-2 spike protein, there were eventually many bound regions reported (also in the Shrock CoV paper). Would those be over the 0.5 threshold, would that even count as a long protein? Why limit to 3 peptides per protein? What if there are 20 super highly predicted, non overlapping motifs in a 300 aa protein? Wouldn't it make sense to pick all? And if there are no peptides with binding predicted at all in a 1000 aa protein, wouldn't it make sense to pick none of them?

• One downside should be mentioned, that eventually it is not possible to know which of the three stitched together proteins were bound...

Point-by-Point responses to the Editor's and Reviewer's comments

Reviewer #1 (Remarks to the Author):

In this study, Liebhoff et al develop an algorithm to improve the efficiency with which large epitope spaces (and specifically, the gut phageome) can be represented in high-dimensional peptide-based serological assays. They use a large experimentally-generated training data set to build a model for linear epitope prediction, and then use this model to generate a library with which they profile anti-phage antibodies in healthy subjects. This is an important area of research given the fact that phage represent a highly-diverse and under-explored space, and the immune response to them has the potential to be a powerful modulator and reporter of the state of the microbiome. However, enthusiasm for the manuscript is reduced by the following concerns that go to the level of innovation, impact and validation of the findings.

We thank the reviewer for this positive comment and agree with the assessment of the importance of this area of research. We made several changes to address the reviewer's comments as detailed below.

1. The authors do not establish the innovation of their RF classifier over the various published epitope prediction approaches. A number of algorithms for the prediction of B cell epitopes (include linear epitopes) from primary amino acid sequence have already been described (e.g. PMIDs: 36366745, 23667458), using increasingly sophisticated machine learning approaches. These existing models are trained on datasets of similar size or larger than the one presented here, and show comparable/superior performance (AUC ~0.6-0.8). The authors do not cite any of these studies, nor attempt to put their findings into that context (eg by comparing the performance their model side-by-side with existing ones).

We agree that citations for these important studies should have been included. In fact, we had tested BepiPred predictions in an earlier version of our classifier but did not find significant improvements for our binary classification of epitopes on 15 amino acid long peptides, so omitted this part for conciseness. We have now added the following content:

This suggests that we can distinguish peptide sequences more likely to contain epitopes by their amino acid composition [...]. Indeed, previous studies (Clifford et al. 2022; Singh, Ansari, and Raghava 2013) have developed models with marginal predictive power.

And further elaborated in the results (section *Binary classification of epitope peptides*):

Furthermore, we explored using protein structures predicted by BepiPred-2.0 (Jespersen et al. 2017) and Reprof (Rost 2003) as features in our model. Including these models did not provide enhancement of classification performance to justify the additional computational costs associated with extracting and incorporating these features.

2. Overall, Dolphyn likely represents only an incremental step towards the authors' stated goal of addressing the vastness of the gut phageome. The authors show that Dolphyn achieves a ~3-fold enrichment of reactive peptides over Pepsyn (Fig 4C): however the majority of this effect (~2X) would be expected from the simple step of eliminating the 2X tiling redundancy of Pepsyn. (It is also important to acknowledge that this 2X compression removes the internal replication that was an intentional part of the original VirScan design approach). Overall, therefore, the impact of the RF classifier therefore appears to be modest, which is consistent with the observed AUCs.

We thank the reviewer for contemplating the utility of our approach in detail. The comment also revealed that we did not sufficiently present all of the pertinent benefits. The three-fold enrichment is over the percentage of reactive peptides. As Pepsyn is not a feasible strategy to address the vastness of the gut phageome, compression is needed. We therefore reduced the redundancy and sketched the proteomic space. However, random sketching would only lead to 10% reactivity. With Dolphyn, we primarily eliminate regions from the non-reactive part of the protein and therefore achieve a much higher percentage of reactive tiles. The reviewer is certainly correct in that the observed AUCs would not suggest dramatic benefit, which we acknowledge in the text. Much of the benefit is also due to the stitching step of our algorithm, since an antibody has three potential targets per library member (peptide).

We attempted to improve clarity of these two points by revising the abstract as follows:

[...] Dolphyn shrinks the size of a peptide library by 78% and increases the percentage of peptides bound by antibodies from 10% to 31% [...]

We also added the following paragraph to the discussion:

The algorithm's stitching step combines three potential epitopes into one library peptide. While the prediction model enables the prioritization of protein regions during compression, the stitching step notably enhances peptide reactivity.

Relatedly, the following conclusion in the abstract "Dolphyn improves the fraction of gut phage library peptides bound by antibodies from 10% to 31% in healthy individuals, while also reducing the number of synthesized peptides by 78%", seems like a case of double-counting. It would be more appropriate to describe these effects disjunctively: a ~3X increase in the proportion of reactive peptides, OR a ~3X reduction in the number of peptides that need to be synthesized to achieve a fixed number of reactive peptides.

We believe the reviewer refers to Figure 3C, in which we present the number of reactive peptides on the y-axis, comparing Pepsyn and Dolphyn library members needed on the x-axis. We attribute this effect solely to the compression algorithm (predicting, sketching and stitching). The fact that we represent the same proteomic space (with less coverage) is stated in the 78% decrease of library size. We hope our new phrasing in the abstract (see above) better reflects these two independent contributions.

3. The profiling of antibody responses to gut phage in healthy subjects (Figs 4,5) is intriguing. The interpretation of these findings, however, is difficult in the absence of any attempt to validate these findings against expected reactivities / orthogonal measures / covariates.

We are aware that our findings open an entirely new opportunity and field of investigation. Unfortunately, our ability to validate bacteriophage reactivity or neutralizing antibody activity is not possible due to the unavailability of cultured bacteriophages represented in our library. The main focus of our article is aimed at the development of the Dolphyn algorithm and the pilot library for gut phages only serves for comparing the former and new approach.

Reviewer #2 (Remarks to the Author):

This manuscript describes the development and application of a new approach and package for designing PhiP-Seq libraries. This package called Dolphyn reduces the number of peptides needed in a library by removing those that don't have amino acid features of epitopes, and then 'stitching together' predicted 15-mer epitopes from the same protein with a GGGGS linker to generate a 56-mer containing 3 15-mers. Using this approach the authors show that at a phage library level the Dolphyn approach reduces library size markedly, and predicts similar peptides compared with the Pepsyn tiling approach. The paper is clear and the rationale and approach is well described.

We thank the reviewer for their time and appreciation of our manuscript.

My major question is around interpretation of findings/epitopes using the new Dolphyn approach - once a 56mer is identified as a possible hit, what should be done next to validate this as a biomarker/public epitope?

The answer to this question depends on the study goal. A "hit" to one of the library members indicates a likely immune response to the corresponding protein that could be validated for example using protein ELISA. In the present study, reactivities to bacteriophage may indicate exposure and potentially neutralizing reactivity. Unfortunately, our ability to validate bacteriophage immunity is not currently possible due to the unavailability of cultured bacteriophages represented in our library.

The 56-mer contains 3x15mer epitopes but the current paper does **not, [sic]** show any post-PhiP-seq analysis or validation to support their selection of the 'most likely' epitope as the 1st 15-mer, followed by the 2nd most likely and then the 3rd.

The selection of 'most likely' epitopes is based on the probability calculated by our random forest model. We clarified this in the manuscript by adjusting the corresponding sentence into:

Specifically, we used the 15-mer with the highest probability determined by the random forest classifier at the first position, [...]

Figure 4B shows that this predicted likelihood does indeed associate with more frequent peptide reactivities.

Is this likely to be confirmed or reflected in peptide ELISA or similar using the 3x 15-mers?

Correct, we expect that peptides with a higher random forest prediction score will be associated with higher seroprevalence (all other factors being equal) as measured by either PhiP-Seq or ELISA.

Or is it suggested that the stitched peptide is used for orthogonal validation?

The stitched peptide is not considered as orthogonal validation in our study.

Have the authors performed any follow on analysis along these lines from the hits they have identified to show the value of Dolphyn beyond PhiP-Seq?

We expect that Dolphyn will for example be broadly useful in vaccine design. However, we have so far only used it for compression of antigenic spaces represented in PhiP-Seq format.

Minor comment - this paper is quite heavy on acronyms and package names. Some could be spelt out in full, or referred to in full at first use, especially for non-bioinformatic audiences eg DIAMONDS, GPD, VRC, PEDS etc

We agree with this suggestion. While DIAMOND is the chosen name by the authors in their publication, we have removed most acronyms throughout the manuscript (GPD, PEDS, RF, PCA, IEDB, OOB) and believe this has improved the clarity and readability of the manuscript.

Reviewer #3 (Remarks to the Author):

Summary

Liebhoff et al. report interesting work that can definitely deserve publication in Nat comms. They have generated relevant new insights not only for researchers using PhIP-Seq, but also more generally into immunology, the nature of antibody bound epitopes etc. The Dolphyn model generated is probably also relevant for researchers designing peptide arrays with short x-mers.

Regarding the presentation, the paper covers two “stories” at once: 1. A new method to reduce the epitope space for antigen library design (called Dolyphn) and 2. studies on anti-gut phage antibody responses. The 1st part feels stronger and is the main focus, the 2nd part lacks key insights into the biology (if there are Igs against the phages, do the abundances of the respective host bacteria change etc.). However, as the title is focused on the 1st part, I do not see an issue with that. The phage antigens can be seen as validation example for the algorithm, and for that they serve their purpose.

The experimental work is extensive, the controls are sound, application of ML nicely carried out with held out controls sets. I see little issues with the experimental work, and my comments below relate mostly to the presentation and interpretation. Since there are quite a few comments, I have recommended major revision, but this does not affect my assessment that the paper is well suited for Nat comms.

We thank the reviewer for their appreciation of our work and recognition for its broad utility in immunobiology. We believe the comments have helped us to greatly improve the overall clarity of the manuscript as well as improved detailed explanations in specific areas.

Major comments

- Some of the more interesting insights in the Dolphyn design are a bit buried in the manuscript (e.g. a key point “This suggests that we can distinguish peptide sequences more likely to contain epitopes by their amino acid composition, which defines the primary and partially secondary structure of proteins.”). This is quite a fundamental insights, which amino acid features are predictive of antibody binding – these findings are not highlighted. The authors may want to highlight this in a more pronounced way.

We agree that this could have been highlighted more clearly and have added the following sentence to the discussion:

Additionally, our results contrast prior work that could not reliably distinguish epitopes based on amino acid sequences alone (Akbar et al. 2021). We are the first to apply this principle to compression of synthetic antigenic spaces. However, our epitope prediction model could also be used in vaccine research or other immunobiology applications.

- The beginning of the Results section details two approaches of profiling the PEDS dataset, 1.) alanine scan and 2.) the k-mer scan. While the alanine scan approach is decently explained, I did not find a clear explanation of the k-mer approach and why it was performed – please add in the main text.

This part was initially left out of the main text due to word-count constraint and was provided in the figure caption only. We added the following two sentences to the results section:

The k-mer scan considers sub-peptides of varying lengths ($k = 15$ to 45 amino acids) derived from the wildtype sequence. They are tiled in steps of five amino acids along the sequence.

- “To assess performance on an independent dataset, we collected a set of Epstein-Barr-Virus (EBV) peptides that had been previously screened in a study by Monaco et al. (2021).”

- o Authors Larman and Elledge have published a lot more datasets. Why not run Dolphyn on more of those as proof of concept? Why exactly on the EBV data? Couldn't you take the full library content of any other paper, take the peptide sequences as input for Dolyphn and compare the prediction outcome to the actual antibody binding signal previously measured? That would strengthen the paper.

We appreciate this excellent suggestion. We chose EBV peptides specifically because we can assume a ground truth - that ~95% of most study populations are EBV seropositive. Including other peptides is

challenging since the absence of reactivity, i.e. no-one in the cohort having antibodies, is not distinguishable from the absence of infection, and therefore the true/false negatives cannot be estimated correctly. However, based on the reviewer's suggestion we expanded this experiment for rhinoviruses and human cytomegalovirus, now reported in new Figure 2D and described in the results section:

To assess performance on an independent dataset, we collected a set of Epstein-Barr virus (EBV) peptides, rhinovirus peptides and human cytomegalovirus peptides that had been previously screened in the VirScan study by Monaco et al. (2021).

- “This suggests that we can distinguish peptide sequences more likely to contain epitopes by their amino acid composition, which defines the primary and partially secondary structure of proteins.”
 - o What about non GRAB motives, in the Shrock 2023 Science the authors state: “*This work has several implications: first, it suggests that private rather than public neutralizing antibodies may be superior candidates for inclusion in therapeutic monoclonal antibody cocktails, because private antibodies are less likely to exert population-wide selective pressures on pathogens and may thus retain efficacy for future variants.*”
- According to that, wouldn't some private epitopes be missed by the training on shared epitopes?

Yes. By training on public/shared epitopes we are not able to make any claims about population-wide predictive features of private epitopes. They may be shared or may be different. However, the primary objective of our study was to reduce the number of non-reactive peptides in an assay, hence prioritizing public epitopes.

- This is a valid way to reduce the epitope space and not against this manuscript here, just mention these considerations to let readers judge if they want to use this or not.

Absolutely! We added clarifying paragraphs (calling it “limitation paragraphs” in this revision) in the discussion, to better describe the limitations of our study. Here is the relevant sentence for this part:

For the purpose of maximizing detection efficiency, we have focused in this study on identifying features of public epitopes. It is important to note that features predictive of private epitope selection may be similar or entirely different.

- The whole paper is not written in the most clear way, and in parts a bit confusing, (such as the bit on k-mers and ala scan mentioned above), if possible streamline a bit, maybe add an overarching figure/schematic that connects everything and add a paragraph in the beginning of the results where you briefly outline what you are doing. I was reading this and not knowing what to expect or why certain things were mentioned at first, later most made sense. Having an outline of what is about to come would have made it easier to read the paper.

Thank you for this constructive feedback. We hope to have improved clarity in the revised version of the manuscript. Instead of a graphical abstract (as Nature Communications does not allow a graphical abstract) we added the following paragraph to the beginning of the results section:

In the following subsections, we analyze epitope characteristics, with a particular emphasis on epitope length and predictive patterns. We then formulate a binary classifier to predict presence of epitopes within 15-amino acid peptides. Next, we introduce the Dolphyn algorithm, which efficiently compresses large sets of proteins into concise peptide libraries. To validate the algorithm, we compare a conventional library design with one produced by Dolphyn. Finally, we construct and evaluate a pilot library encompassing gut phage proteins, employing and comparing the traditional and novel library design methods.

- “Simple tiling methods like Pepsyn (Larman et al. 2011) divide a protein into peptides of equal length with some overlap (Fig 3A). This approach wastes resources on synthesizing, cloning, and sequencing peptides that are not reactive.”
 - o Not sure I agree, related to above comment. Would antibody bound GRAB and none. GRAB motives be equally well predicted? The notion of the statement above (if I understand the process correctly applies mostly to GRAB motifs, what is the overall ratio of GRAB vs. non-GRAB antibody bound motifs? Discuss briefly.

Thank you this comment, which made us realize the need for a clearer demarcation between the discussions related to Figure 2 (GRAB motifs) and the commencement of the results presented in Figure 3. The specific sentence mentioned, "Simple tiling methods...", serves the purpose of re-emphasizing the requirement for a more efficient library creation algorithm. It's important to note that preceding this sentence, we had discussed GRAB motifs as a means of validating our findings regarding significant features within epitopes. If the inquiry pertains to the frequency of GRAB motifs occurring along a protein, it is essential to acknowledge that GRAB motifs lack sufficient specification to conduct a targeted search for them (discussed in Shrock et al. 2023).

We have taken your feedback into consideration and made adjustments to the paragraph in the manuscript to ensure a clear distinction between the overarching concept and the discussion of GRAB motifs as separate topics:

Simple tiling methods [...] with some overlap (Fig. 3A). This approach has been used to design libraries shown in Figure 1A and wastes resources on synthesizing, cloning, and sequencing peptides that have a very low probability to ever be reactive in anyone.

- The paper focuses heavily on the benefits of Dolphyn, but does not highlight the limitations:
 - o "In order to detect the same number of immune responses as a Dolphyn library, a Pepsyn library requires about three times the number of peptides."

Thank you, we have adjusted the following points accordingly.

- It needs to be mentioned how many signals would be missed (false negatives in a way) using the Dolphyn approach.
 - o In other words, in "Reactivity of Dolphyn libraries" the absolute number of bound peptides need to be mentioned, how many more were only detected with regular tiling, than with the prediction and stitching approach?

To address this comment we added the absolute numbers of reactive peptides to the mentioned paragraph: *Dolphyn-designed libraries demonstrate similar discovery power (accuracy) for identifying protein antibody targets in 53 individuals, as shown in Figure 5D. Dolphyn peptides only recall about a third of the proteins (131) that are reactive in Pepsyn (469) partially because Dolphyn does not include some proteins.*

It should also be mentioned that assays aren't perfect and that Dolphyn found proteins to be reactive which were not detected by regular tiling with Pepsyn. Throughout the manuscript we treat the regular tiling as "gold standard" and evaluate Dolphyn against this in Figure 5D. The evaluation for treating Dolphyn and Pepsyn hits as the combined ground truth can be found in the Supplementary Material (Protein Discovery Power). Please note also that we found a versioning error thanks to this comment, which led us to correct the numbers and figure.

- o I think this comes up later in a way "Dolphyn peptides only recall a bit more than half the proteins that are reactive in Pepsyn partially because Dolphyn does not include some proteins." But should be mentioned first in the relevant section where readers would look for this info.

We added a paragraph where Dolphyn is introduced for the first time, referring to the numbers added in the previous point:

Since the goal of Dolphyn is to represent a smaller, more reactive portion of the original proteome, some epitopes may be missed, especially when there are multiple epitopes located at overlapping positions, or when there are fewer than three epitopes predicted on a protein. Furthermore, because Dolphyn stitches together peptides from different locations across a protein, reactivity data must be interpreted at the protein level.

We agree that the placement of this explanation should be earlier in the manuscript and hope this new paragraph improves clarity for the reader. Please note that the comparison of Pepsyn and Dolphyn is only possible at the protein level. Peptides are not mappable one-to-one and therefore there is not a straightforward answer to the question which epitopes were missed by Dolphyn.

o “The algorithm reduces the number of peptides in a library for a given proteome by 78% and triples the reactivity per peptide, as compared to uniform protein tiling”

- This should followed with a sentence like “Compared to the reactivities in the x-fold more extensive/variant intensive conventional tiling, Dolphyn misses approx. half the proteins of the conventional tiling, because they are simply not included in the Dolphyn library.” (you can sure write this in a better way)

We adjusted the manuscript related to this suggestion accordingly:

The algorithm [...] as compared to uniform protein tiling with Pepsyn. Dolphyn peptides recall 29% of the gut phage proteins that exhibited reactivity with Pepsyn designed peptide tiles. Some of the lost recall is attributed to Dolphyn's exclusion of specific proteins from its representation (e.g. proteins with fewer than three predicted epitopes).

- Here it should also be highlighted, that there is hence a tradeoff between false negatives (missed ones, because they were simply not included) using this approach – if one is interested in a particular species or set of antigens, it may be better to represent them by full tiling, rather than Dolphyn.

This is correct. It may often be the case that an investigator prefers to maximize sensitivity and mappability at the expense of encoding efficiency, for small and tractable antigenic spaces. We address this now in the “limitation paragraphs” in the discussion section with the following sentence:

Our algorithm enables the study of immune responses to very large protein sets. [...] The trade-off for a compressed library design also reduces resolution. Utilizing Dolphyn libraries, we explored general antibody reactivity to proteins or species rather than identifying specific epitopes. For optimally identifying epitopes in a reduced set of species, an enhanced tiling approach can be employed as recently demonstrated by (Na 2022).

• Key limitation to mentioned: with 50 to 90 aa peptides at least some secondary structures and semi-conformational epitopes would be represented, with 15 mers only linear epitopes.

Thank you for highlighting this aspect. We included the following sentence in the new “limitation paragraphs” in the discussion section:

For the purpose of maximizing detection efficiency, we have focused in this study on identifying features of linear public epitopes.

Minor comments

• How large is their library, how large cohort?

We have described the library size in the text in section “Compressing the gut phage database for antibody profiling” and in the table. Cohort sizes are given with the respective results (e.g. in “Reactivity of Dolphyn libraries”). We have also added to the table description the following:

The Pilot Library was synthesized and evaluated in this study.

• What about reactivity against T7 phages? Those antigens could not be included as part of the library, on the other hand there may be some antibody responses against T7 phages, that would react against the phage library independently of the antigen displayed on a phage's surface. That would potentially increase background noise and result in low level binding against the entire library – discuss implicaitons/mention this issue.

The reviewer makes a good point and we have added this consideration to the limitation paragraphs:

Additionally, it should be acknowledged that anti-T7 phage antibodies, if present, might introduce background noise to assay results. To enhance the sensitivity of future assays, we recommend considering the incorporation of a PCR spike-in control, which could facilitate the detection and quantification of this library-wide background noise.

• P. 3 “a cohort of 59 individuals (425 samples)” – what done with multiple samples per person?

Correct, in the methods we describe that there are several samples per person (from various time-points) and added an explanation:

Response by Liebhoff et al. - Dolphyn Manuscript for Nature Communications

For the Public Epitope Data Set samples were treated independently, as individual antibody responses change over time.

However, for calling a peptide “reactive”, two or more samples must have shown reactivity, as described in “Computational Methods - ML training set”, to exclude spurious false positives from analysis.

- Anti-gut phage antibody responses have been measured with PhIP-Seq in these papers, discuss briefly (the info on the phages is there somewhat scarce, but they show up in the associations etc. reported): PMIDs 37164013 and 37164015

We appreciate the suggestion to examine these recent publications, which overlapped with our submission process. We included their citations as part of the introduction as follows:

Recent data (Andreu-Sánchez et al. 2023) support the presence of immune responses to gut phages, along with the importance of anti-gut phage antibodies in inflammatory bowel disease (Bourgonje et al. 2023).

We further included the phage library of *Leviatan et al.*, that these studies were based on in our list of references.

- Public Epitope Data Set (PEDS) – is this available as supporting table/xlsx/csv file? Should be added or explain why not added.

Because of its size, the PEDS is not available on GitHub but on Zenodo instead (<https://doi.org/10.5281/zenodo.7979557>) The GitHub repo contains the information how to integrate/download the data from Zenodo. The Data Availability section contains this information. We have added the following phrase when introducing the Public Epitope Dataset:

The resulting Public Epitope Dataset profiles [...] (Methods and Figure 1B, data available).

- Methods:

- o Public Epitope Library – this is not linked to the Shrock 2023 Science paper? Because no reference to that paper is given.

The library was designed by Ben Larman and Tomasz Kula in Steve Elledge’s lab and is available to both laboratories. We added a short statement to the methods text for the attentive reader:

This library [...], and has also been used by Shrock et al. 2023 with a different cohort.

- o “The final Dolphyn library is comprised of 48,128 individual 200-mers.”
Maybe start out with that first, and then explain that sub libraries.

Thank you for this suggestion. We removed the sentence and adjusted the first sentence in the methods text about the Dolphyn/Phageome library accordingly:

This library contains 48,128 peptides that are 56 amino acids long and is divided in three subsets of peptides, representing the same 112 prevalent phages in 3,354 protein cluster representatives: [...]

- o Enterovirus Sublibrary: What’s the total number of variants here?

We used seven (7) reference enterovirus sequences (described in the text).

- o Did you observe any differences in the quality / error rates / uniformity between the library synthesized by Twist Bioscience (San Francisco, CA) and the other one by GenScript Biotech (Piscataway, NJ)? Could there be any bias affecting the results?

GenScript libraries have higher error rates compared to Twist and they cannot make oligos of the same length (due to this limitation). However, they provide a faster turnaround time and more favorable pricing in comparison.

- Author list order in the manuscript and online system is different – on purpose?

Thank you for catching this detail. The order on the manuscript is the correct one.

The order is the same in the online system except for the fact that the corresponding author (Ben Larman) had to be entered first, even though he is the last author on the manuscript.

- “For each protein, Dolphyn selects the three non-overlapping 15-mers with the highest epitope probability (Fig. S1). For long proteins, multiple sets of three 15-mers are selected if the probability of containing an epitope is greater than 0.5.”

o Discuss why this approach was chosen.

We agree that the description of our approach is somewhat condensed. We elaborate below on the description in the text.

- **We address each protein and determine the probability to contain an epitope for each possible sub-15-mer exhaustively**
- **Each 15-mer has a probability to contain an epitope. If this probability is greater than 0.5, we call it “peptide/15-mer probably containing an epitope”**
- **Depending on how many 15-mers were predicted to contain epitopes, Dolphyn estimates the amount of stitched peptides to be made: If 3-5 epitopes were predicted -> one stitched peptide is made, 6-8 -> 2 stitched peptides and so on.**
- **Here, we potentially “throw away” 1 or 2 potential epitopes for the sake of efficiency. We don’t want to “waste” open positions on stitched peptides just for making a peptide that contains very low probability epitopes.**

We also improved the description of the algorithm (see the following two points below).

- What is a “long” protein? Say you take the SARS-CoV-2 spike protein, there were eventually many bound regions reported (also in the Shrock CoV paper).

In this case a “long protein” is one that contains six or more predicted epitopes (so that two or more stitched peptides can be made). Approximately one epitope per 45 amino acids is predicted, which results in about 270 amino acids for a “long protein”.

- Would those be over the 0.5 threshold, would that even count as a long protein?

The threshold of 0.5 is the prediction value that the Random Forest classifier generates per 15-mer. It is independent of the length of the protein. We realized that the paragraph with the algorithm description provides insufficient information, so we have revised the paragraph as follows:

Dolphyn predicts whether each 15-amino-acid sub-peptide (15-mer) of a protein contains an epitope using the random forest model described above that was trained on our public epitope dataset (Fig. 2). For each protein, depending on its length, Dolphyn selects a multiple of three non-overlapping 15-mers with the highest epitope probability (Fig. S1). 15-mers are selected if their probability of containing an epitope is greater than 0.5. Then, sets of three 15-mers are combined using Dolphyn’s “stitching” step and separated via a flexible and inert linker sequence GGGGS (see Fig. S2 for evaluation of different linkers). The resulting peptides are used to evaluate antibody reactivity at the protein level. A pseudo-code description of this algorithm is given in Figure S1 section.

We added a pseudo-code description to the supplementary material:

Dolphyn ()

```
for each protein_sequence:

    //PREDICTION STEP
    for each 15mer in 15-amino-acid-sub-peptide:
        probabilities[15mer] = contains_epitope(15mer) // C1
    threshold = 0.5
    epitope_15mers = select_non_overlapping(probabilities > threshold) // C2

    //STITCHING STEP
    np = number_of_peptides = length(epitope_15mers) mod 3 // C3
    for i in 0 to number_of_peptides:
        library_peptide[i] = epitope_15mers[i] + GGGGS
        linker_seq = GGGGS // C4
        for i in 1 to number_of_peptides:
            library_peptide[i] = epitope_15mers[i] + linker_seq +
                epitope_15mers[i+ np] + linker_seq +
                epitope_15mers[i+ (np*2)] + stop_codon //C5

//COMMENTS
C1: with random forest model, trained on public epitope dataset (Fig2)
C2: epitope_15mers ordered by probability
C3: if 2 or less epitopes predicted: no library peptide made,
for 3-5 epitopes -> one peptide, for 6-8 epitopes -> two peptides, etc
C4: see Fig S2 for evaluation of different linkers
C5: this puts highest probability 15mer on 1st position of 1st peptide
and 2nd highest probability on 1st position of 2nd peptide etc
```

- Why limit to 3 peptides per protein? What if there are 20 super highly predicted, non overlapping motifs in a 300 aa protein? Wouldn't it make sense to pick all?

The limitation is not on 3 peptides per protein but 3 epitopes per peptide. In the given example it is correct that the 18 epitope-peptides (15-mers) with the highest prediction values will be chosen to make 6 stitched peptides. The last two will be left out for efficiency reasons, but their prediction value would be closer to 0.5 (and not “highly predicted” anyways). If it were 20 “highly predicted” epitope-peptides, there could also be a 21st epitope with a lower prediction value which would then be used to make a seventh tile.

- And if there are no peptides with binding predicted at all in a 1000 aa protein, wouldn't it make sense to pick none of them?

In the very unlikely case that two or fewer peptides are predicted in a 1000 amino acid protein, indeed no 15-mer would be picked, and no representative Dolphyn peptide would be made.

• One downside should be mentioned, that eventually it is not possible to know which of the three stitched together proteins were bound...

That is correct. We included this point in combination with another reviewer's question in the “limitation paragraphs” in the discussion section:

The trade-off for a compressed library design is reduced antibody binding resolution. Utilizing Dolphyn libraries, we explored general antibody reactivity to proteins or species rather than identifying specific epitopes.

Reviewer #1 (Remarks to the Author):

The authors have made good improvements to their manuscript in response to many of the reviewers comments, including mine. However, their response to my comment #1 (regarding how their RF classifier compares with the prior art) remains unsatisfactory.

What I think is needed here is a comparison of how the performance of the new method compares with these prior approaches. The authors offer a new comment that incorporating the prior approaches "provided insufficient enhancement of classification performance" (without showing any supporting data). Rather than showing superiority over the prior approaches, this may actually suggest redundancy with them.

A more informative analysis would be to run both methods (e.g. Dolphyn v BepiPred), side-by-side on the same "test" data and compare their performance. In the Introduction, the authors describe the existing models as having "marginal predictive power" – by this definition, is the performance of the authors' RF (whose AUCs appear similar) also marginal?

The following claim in the Discussion also needs more complete substantiation against the prior art: "our results contrast with prior work that could not reliably distinguish epitopes based on amino acid sequences alone".

Without a proper analysis of the prior epitope prediction approaches, it is hard to assess the impact/significance of the new method, or the degree to which these claims are supported.

Reviewer #2 (Remarks to the Author):

The authors have prepared a comprehensive rebuttal and answers all of my minor queries. However my major comment on orthogonal validation of the findings (also asked by another reviewer) has not been fully addressed.

A key feature of this new library design is to stitch 3 15-mer peptides (likely epitopes) together to compress library size. However, none of the 'stitched' peptides identified have been orthogonally validated, so the real benefits of this approach in terms of epitope discovery are not fully supported with the current data.

The response is: "Unfortunately, our ability to validate bacteriophage reactivity or neutralizing antibody activity is not possible due to the unavailability of cultured bacteriophages represented in our library."

However, not having the phage shouldn't negate the ability to validate - the 3x15-mer peptides could be quickly synthesized from a commercial provider and elisa performed. This would experimentally show that the peptides are hits and also be informative as to whether their predictions of most likely epitopes (where these were placed in order 1, 2 and 3 in the "stitched peptides") are correct.

I think it would significantly strengthen the current study to include validation along these lines, even if just for one or two targets as an example.

Reviewer #3 (Remarks to the Author):

The authors have diligently replied to all my comments, only minor points should be addressed:

The whole discussion around -non GRAB motifs:

Old text: "This suggests that we can distinguish peptide sequences more likely to contain epitopes by their amino acid composition, which defines the primary and partially secondary structure of proteins."

o What about non GRAB motifs, in the Shrock 2023 Science the authors state: "This work has several implications: first, it suggests that private rather than public neutralizing antibodies may be superior candidates for inclusion in therapeutic monoclonal antibody cocktails, because private antibodies are less likely to exert population-wide selective pressures on pathogens and may thus retain efficacy for future variants."

- According to that, wouldn't some private epitopes be missed by the training on shared epitopes? Yes. By training on public/shared epitopes we are not able to make any claims about population-wide predictive features of private epitopes. They may be shared or may be different. However, the primary objective of our study was to reduce the number of non-reactive peptides in an assay, hence prioritizing public epitopes."

 Please add a short explanation/summary of these issues to make readers also aware of this (describe the Shrock work, potential issues with private epitopes, and what Dolphyn is aimed at). Add this please even beyond what you have added now ("For the purpose of maximizing detection efficiency, we have focused in this study on identifying features of public epitopes. It is important to note that features predictive of private epitope selection may be similar or entirely different.")

Another similar point:

Old text: "- Here it should also be highlighted, that there is hence a tradeoff between false negatives (missed ones, because they were simply not included) using this approach – if one is interested in a particular species or set of antigens, it may be better to represent them by full tiling, rather than Dolphyn.

This is correct. It may often be the case that an investigator prefers to maximize sensitivity and mappability at the expense of encoding efficiency, for small and tractable antigenic spaces. We address this now in the "limitation paragraphs" in the discussion section with the following sentence:

Our algorithm enables the study of immune responses to very large protein sets. [...] The trade-off for a compressed library design also reduces resolution. Utilizing Dolphyn libraries, we explored general antibody reactivity to proteins or species rather than identifying specific epitopes. For optimally identifying epitopes in a reduced set of species, an enhanced tiling approach can be employed as recently demonstrated by (Na 2022)."

 I think the explanation to the reviewer ("It may often be the case that an investigator prefers to maximize sensitivity and mappability at the expense of encoding efficiency, for small and tractable antigenic spaces.") is more clear than what was added to the discussion section. Please add that sentence also to the paper.

REVIEWER COMMENTS

Reviewer #1 (Remarks to the Author):

The authors have made good improvements to their manuscript in response to many of the reviewers comments, including mine. However, their response to my comment #1 (regarding how their RF classifier compares with the prior art) remains unsatisfactory.

What I think is needed here is a comparison of how the performance of the new method compares with these prior approaches. The authors offer a new comment that incorporating the prior approaches "provided insufficient enhancement of classification performance" (without showing any supporting data). Rather than showing superiority over the prior approaches, this may actually suggest redundancy with them.

A more informative analysis would be to run both methods (e.g. Dolphyn v BepiPred), side-by-side on the same "test" data and compare their performance.

We agree and thank the reviewer for this suggestion. We added such an experiment to the manuscript. We ran the latest version of BepiPred on our validation set of reactive and non-reactive 15-mers and conducted two experiments. First, we gave the validation set 15-mers as inputs to BepiPred and averaged the epitope scores for all 15 residues, allowing the 15-mers to be prioritized in a similar manner as Dolphyn does. The resulting AUC was 0.54, lower than Dolphyn's AUC of 0.7. Second, rather than giving the isolated 15-mers as input, we provided longer protein sequences to BepiPred, thus providing more sequence context than either BepiPred or Dolphyn had been given in the first experiment. We still averaged the residue-by-residue predictions from BepiPred within the 15-mer windows to obtain a final score for tile prioritization. Under these conditions, BepiPred's AUC was 0.74 which was slightly higher than Dolphyn's AUC of 0.7. This result suggests that BepiPred's predictions are of lower utility when applied in the exact same context as our RF model, but of comparable utility when given wider context about the protein. It is thus fair to say that we do not "surpass" BepiPred in any substantial way; rather, we are adapting it to our design problem. Overall, the experiments support our contention that Dolphyn's approach is a particularly effective approach for our chosen PhIP-Seq tile design problem, even if it is not the only viable approach.

We would like to emphasize that we did not aim to outperform existing epitope prediction models, but rather to develop an algorithm serving our specific application. Dolphyn's random-forest approach is (a) computationally inexpensive, (b) conceptually simple, being based only on amino acid frequencies, and (c) tailored to our specific classification question and training data, which is concerned specifically with determining whether a candidate PhIP-Seq peptide tile likely includes a public epitope. Residue-by-residue predictors like BepiPred are certainly related to our application, but would require some nuanced adaptation before they would work in the context of our training data and classification question. We updated Figure 2E accordingly and added in the results section:

For the sake of comparison with an independent epitope prediction tool, we also determined the average BepiPred-3.0 score (Clifford et al. 2022) for the probability of a 15-mer containing an epitope. Depending on the amount of data given to the algorithm as input, the resulting ROC curves have an AUC ranging from 0.54 to 0.74. The lower score is based on the same input data that is used for the random forest;

the higher score results from having the surrounding protein sequence inform the prediction. These results indicate that other algorithms may be comparable in performance, but would likely involve further adaptation for use in our specific application.

In the Methods section we added:

BepiPred Epitope Prediction on Enterovirus sublibrary peptides

For the ROC curve generated in Figure 2E, BepePred-3.0 was run with default settings through <https://biolib.com/DTU/BepiPred-3> in two experiments. The first included the 15 amino acid long sequences of the Enterovirus sublibrary. The second experiment used the seven reference enterovirus sequences as input. In both experiments, the BepiPred scores for an epitope at each residue were averaged over the 15 amino acids contributing to a peptide. For creating the ROC curve, the same labels were used as in the validation of the random forest, derived from the Enterovirus library screening cohort.

In the Introduction, the authors describe the existing models as having "marginal predictive power" – by this definition, is the performance of the authors' RF (whose AUCs appear similar) also marginal?

We agree that this was a misleading way to phrase the statement. We have revised the sentence to read:

*Indeed, previous studies (Clifford et al. 2022; Singh, Ansari, and Raghava 2013) **have also developed models to achieve this goal.***

The following claim in the Discussion also needs more complete substantiation against the prior art: "our results contrast with prior work that could not reliably distinguish epitopes based on amino acid sequences alone".

We thank the reviewer for highlighting this reference which paraphrased the author's original statement:

"For epitopes, several analyses have shown that their amino acid composition is essentially indistinguishable from that of other surface-exposed non-epitope residues if the corresponding antibody is not taken into account."

[Akbar, Rahmad, ..., Cédric R. Weber, et al. 2021. "A Compact Vocabulary of Paratope-Epitope Interactions Enables Predictability of Antibody-Antigen Binding." Cell Reports]

While we had previously focused on the aspect of predicting epitopes without prior knowledge of the corresponding antibody, we have now re-considered our interpretation of the authors' statement, which is specifically related to surface-exposed residues. We therefore decided to remove this reference from the manuscript.

~~Additionally, our results contrast with prior work that could not reliably distinguish epitopes based on amino acid sequences alone (Akbar et al. 2021).~~

Without a proper analysis of the prior epitope prediction approaches, it is hard to assess the impact/significance of the new method, or the degree to which these claims are supported.

We would like to thank the reviewer for insisting on a fair comparison with previous methods, as it has added important nuance to our analysis and manuscript. However, we would like to stress that our method is primarily a novel *application* of the kind of predictions that BepiPred and our RF model both accomplish. BepiPred (and others) are not tools specifically for choosing tiles or designing assays, as is

the case for Dolphyn's random forest model. Our aim is not to be better than BepiPred at BepiPred's prediction task, but rather to adapt the same kind of thinking behind BepiPred to our specific problem of prioritizing protein sequence sections to maximize the efficiency of a peptide library design.

Reviewer #2 (Remarks to the Author):

The authors have prepared a comprehensive rebuttal and answers all of my minor queries.

However, my major comment on orthogonal validation of the findings (also asked by another reviewer) has not been fully addressed.

A key feature of this new library design is to stitch 3 15-mer peptides (likely epitopes) together to compress library size. However, none of the 'stitched' peptides identified have been orthogonally validated, so the real benefits of this approach in terms of epitope discovery are not fully supported with the current data.

The response is: "Unfortunately, our ability to validate bacteriophage reactivity or neutralizing antibody activity is not possible due to the unavailability of cultured bacteriophages represented in our library."

However, not having the phage shouldn't negate the ability to validate - the 3x15-mer peptides could be quickly synthesized from a commercial provider and elisa performed.

This would experimentally show that the peptides are hits and also be informative as to whether their predictions of most likely epitopes (where these were placed in order 1, 2 and 3 in the "stitched peptides") are correct.

I think it would significantly strengthen the current study to include validation along these lines, even if just for one or two targets as an example.

We would like to thank the reviewer for raising the question of orthogonal assay validation. To this end, we would like to highlight two previous publications that demonstrate the concordance between PhIP-data and peptide ELISA data:

1. Venkataraman, Thiagarajan, et al. "Comprehensive profiling of antibody responses to the human anellome using programmable phage display." *Cell Reports* 41.12 (2022).
2. Grant-McAuley, Wendy, et al. "Evaluation of multi-assay algorithms for cross-sectional HIV incidence estimation in settings with universal antiretroviral treatment." *BMC infectious diseases* 22.1 (2022): 1-11.

We added a statement and these references in the introduction accordingly:

Previous publications demonstrate the concordance between PhIP-Seq data and peptide ELISA data (Venkataraman, Swaminathan, et al. 2022; Grant-McAuley et al. 2022).

If instead the reviewer would like us to validate the phage displayed peptides specifically, we would like to address this in the answers below.

We had shown summarized information in Fig 5D (how much reactivity do the stitched peptides recall from another library design (Pepsyn)) and 5E (do the overall response of the stitched peptides reflect the same as the independent Pepsyn design). We now add an analysis to the Supplementary Material

that uses the library of unstitched peptides to report the recall of the stitched peptides which are reactive in several individuals:

Figure S5. Validation of Dolphyn peptides

The predicted epitope 15mers were synthesized in a separate library to the Dolphyn stitched peptide library. In a paired t-test, the null hypothesis was rejected with a p-value of 0.0015 that there is no relation between the reactivity in a sample between the two libraries (Fig S5A). As shown in Fig S4, not all reactive individual peptides are also reactive in the stitched version, or the other way round. Fig S5B visualizes the Pearson correlation for the reactivity of all confirmed peptides.

We would also like to highlight Figure 4A, where we displayed the 15mers as individual peptides and additionally their stitched version. In a column of Fig 4A (copied below for reference), individual 15mers and their combined version is displayed. In most cases, when there is one of the three individual peptides reactive, the stitched version (triplet) is also reactive. As the stitched peptides and the individual 15mers are independent peptides in the library, we consider them to be somewhat of a validation for each other.

Further details: This figure 4A displays all sets of peptides where at least two peptides are reactive. The PhIP-Seq assay is not perfect, visible in the none-reactive triplets in individuals 1 and 4, where two 15-mers are reactive but the reactivity could not be recovered in the stitched version. Accordingly, there are also unvalidated individual 15-mers as well as triplets, which we now added to the Supplementary Material exemplarily for individual 2:

Figure S4. All reactive peptides in an individual

Figure 4A in the manuscript shows for four individuals all peptide quartets (three 15-mers and their stitched version) where two or more peptides show reactivity. Here, exemplary for individual 2, all reactive peptides are shown in the same format. Not all reactive individual peptides are also reactive in the stitched version, and there are stitched peptides that could not be confirmed by any individual epitope.

On prediction performance and peptide ranking: Figure 4B uses the individual 15-mers that are also shown in Figure 4A, independently of their prediction score. They are grouped by whether they were placed first, second, or third position on the stitched version. The figure shows that the first group is the most reactive by itself, as expected since they have the highest prediction scores throughout all proteins.

We also added the following paragraph to the manuscript (Results) for further clarification to the reader:

Figure S4 shows all the reactive peptides for one representative sample in the same style as Figure 4A. Figure S5 indicates how the stitched peptides were validated by the individual epitope library with a p-value of 0.0015.

Reviewer #3 (Remarks to the Author):

The authors have diligently replied to all my comments, only minor points should be addressed:

The whole discussion around -non GRAB motifs:

Old text: "This suggests that we can distinguish peptide sequences more likely to contain epitopes by their amino acid composition, which defines the primary and partially secondary structure of proteins."

o What about non GRAB motifs, in the Shrock 2023 Science the authors state: "This work has several implications: first, it suggests that private rather than public neutralizing antibodies may be superior candidates for inclusion in therapeutic monoclonal antibody cocktails, because private antibodies are less likely to exert population-wide selective pressures on pathogens and may thus retain efficacy for future variants."

- According to that, wouldn't some private epitopes be missed by the training on shared epitopes?

Yes. By training on public/shared epitopes we are not able to make any claims about population-wide predictive features of private epitopes. They may be shared or may be different. However, the primary objective of our study was to reduce the number of non-reactive peptides in an assay, hence prioritizing public epitopes."

 Please add a short explanation/summary of these issues to make readers also aware of this (describe the Shrock work, potential issues with private epitopes, and what Dolphyn is aimed at). Add this please even beyond what you have added now ("For the purpose of maximizing detection efficiency, we have focused in this study on identifying features of public epitopes. It is important to note that features predictive of private epitope selection may be similar or entirely different.")

Thank you for requesting additional clarification about the relevance of the Shrock paper.

- Here we describe the key Shrock findings but are open to additional suggestions:
 - In the Introduction:
However, it has been recently reported that public epitopes tend to contain amino acid sequence features that are important for interactions with germline-encoded antibody domains (Shrock et al. 2023).
 - In the Result section: *Interestingly, a recent study also found lysine to be an important feature of epitopes. A germline encoded feature of antibodies called the "GRAB" motif was described as playing an important role in recognizing public epitopes (Shrock et al. 2023). In humans, these epitopes enrich lysines on their borders if recognized by antibodies using a lambda light chain.*
- potential issues with private epitopes
 - We have clarified that private epitopes are not as useful for compressing a library since including them would by definition only detect a small number of individual responders: *We have focused in this study on identifying features of public epitopes, which would require the fewest number of peptides to detect the greatest number of individual immune responses. Notably, including private epitopes would by definition detect only very few individual responders per library member.*
- what Dolphyn is aimed at
 - We revised the wording of the sentence mentioned by the reviewer: *Our algorithm is aimed at maximizing detection efficiency of immune responses to very large protein sets.*

Another similar point:

Old text: "- Here it should also be highlighted, that there is hence a tradeoff between false negatives (missed ones, because they were simply not included) using this approach – if one is interested in a particular species or set of antigens, it may be better to represent them by full tiling, rather than Dolphyn.

This is correct. It may often be the case that an investigator prefers to maximize sensitivity and mappability at the expense of encoding efficiency, for small and tractable antigenic spaces. We

address this now in the “limitation paragraphs” in the discussion section with the following sentence:

Our algorithm enables the study of immune responses to very large protein sets. [...] The trade-off for a compressed library design also reduces resolution. Utilizing Dolphyn libraries, we explored general antibody reactivity to proteins or species rather than identifying specific epitopes. For optimally identifying epitopes in a reduced set of species, an enhanced tiling approach can be employed as recently demonstrated by (Na 2022).“

 I think the explanation to the reviewer (“It may often be the case that an investigator prefers to maximize sensitivity and mappability at the expense of encoding efficiency, for small and tractable antigenic spaces.”) is more clear than what was added to the discussion section. Please add that sentence also to the paper.

We embedded the desired sentence into the given paragraph as follows:

The trade-off for a compressed library design is reduced antibody binding resolution. Utilizing Dolphyn libraries, we explored general antibody reactivity to proteins or species rather than identifying specific epitopes. In case an investigator prefers to maximize sensitivity and mappability at the expense of encoding efficiency, an enhanced tiling approach can be employed, as recently demonstrated by (Na 2022), which targets small and tractable antigenic spaces, for example a reduced set of viral species.

Reviewer #1 (Remarks to the Author):

The authors have addressed my remaining concerns and added data that I think strengthens the manuscript.

Reviewer #2 (Remarks to the Author):

The authors have now referenced prior studies that have validated Phip-seq findings by ELISA and more clearly showed how their Dolphyn identified 'triple peptides' compared with the 3x15mers from prior studies. The overlap between the 2 approaches serves as somewhat of a validation of the new approach. As such my comments have been addressed

Reviewer #3 (Remarks to the Author):

I only had minor comments in the last version that now have also been addressed.